# XLA: A ROBUST UNSUPERVISED DATA AUGMENTATION FRAMEWORK FOR CROSS-LINGUAL NLP

## Abstract

Transfer learning has yielded state-of-the-art (SoTA) results in many supervised NLP tasks. However, annotated data for every target task in every target language is rare, especially for low-resource languages. We propose XLA, a novel data augmentation framework for self-supervised learning in zero-resource transfer learning scenarios. In particular, XLA aims to solve cross-lingual adaptation problems from a source language task distribution to an unknown target language task distribution, assuming no training label in the target language task. At its core, XLA performs simultaneous self-training with data augmentation and unsupervised sample selection. To show its effectiveness, we conduct extensive experiments on zero-resource cross-lingual transfer tasks for Named Entity Recognition (NER), Natural Language Inference (NLI) and paraphrase identification on Paraphrase Adversaries from Word Scrambling (PAWS). XLA achieves SoTA results in all the tasks, outperforming the baselines by a good margin. With an in-depth framework dissection, we demonstrate the cumulative contributions of different components to XLA's success.

## 1 INTRODUCTION

Self-supervised learning in the form of pretrained language models (LM) has been the driving force in developing state-of-the-art natural language processing (NLP) systems in recent years. These methods typically follow two basic steps, where a *supervised* task-specific fine-tuning follows a large-scale LM pretraining (Devlin et al., 2019; Radford et al., 2019). However, getting annotated data for every target task in every target language is difficult, especially for low-resource languages.

Recently, the *pretrain-finetune* paradigm has also been extended to multi-lingual setups to train effective multi-lingual models that can be used for *zero-shot* cross-lingual transfer. Jointly trained deep contextualized multi-lingual LMs like mBERT (Devlin et al., 2019) and XLM-R (Conneau et al., 2020) coupled with supervised fine-tuning in the source language have been quite successful in transferring linguistic and task knowledge from one language to another without using any task label in the target language. The joint pretraining with multiple languages allows these models to generalize across languages. Despite their effectiveness, recent studies (Pires et al., 2019; K et al., 2020) have also highlighted one crucial limiting factor for successful cross-lingual transfer. They all agree that the cross-lingual generalization ability of the model is limited by the (lack of) structural similarity between the source and target languages. For example, for transferring mBERT from English, K et al. (2020) report about 23.6% accuracy drop in Hindi (structurally dissimilar) compared to 9% drop in Spanish (structurally similar) in cross-lingual natural language inference (XNLI). The difficulty level of transfer is further exacerbated if the (dissimilar) target language is low-resourced, as the joint pretraining step may not have seen many instances from this language in the first place. In our experiments (§4.2), in cross-lingual NER (XNER), we report F1 reductions of 28.3% in Urdu and 30.4% in Burmese for XLM-R, which is trained on a much larger multi-lingual dataset than mBERT.

One attractive way to improve cross-lingual generalization is to perform *data augmentation* (Simard et al., 1998), and train the model (*e.g.,* XLM-R) on examples that are similar but different from the labeled data in the source language. Formalized by the Vicinal Risk Minimization (VRM) principle (Chapelle et al., 2001), such data augmentation methods have shown impressive results recently in computer vision (Zhang et al., 2018; Berthelot et al., 2019; Li et al., 2020a). These methods enlarge the support of the training distribution by generating *new* data points from a *vicinity distribution* around each training example. For images, the vicinity of a training image can be defined by a set

of operations like rotation and scaling, or by linear mixtures of features and labels (Zhang et al., 2018). However, when it comes to text, such methods have rarely been successful. The main reason is that unlike images, linguistic units (*e.g.,* words, phrases) are discrete and a smooth change in their embeddings may not result in a plausible linguistic unit that has similar meanings.

In NLP, the most successful data augmentation method has so far been *back-translation* (Sennrich et al., 2016) which generates paraphrases of an input sentence through round-trip translations. However, it requires parallel data to train effective machine translation systems, acquiring which can be more expensive for low-resource languages than annotating the target language data with task labels. Furthermore, back-translation is only applicable in a supervised setup and to tasks where it is possible to find the alignments between the original labeled entities and the back-translated entities, such as in question answering (Yu et al., 2018; Dong et al., 2017).

In this work, we propose XLA, a robust unsupervised **cross-**lingual **a**ugmentation framework for improving cross-lingual generalization of multilingual LMs. XLA augments data from the unlabeled training examples in the target language as well as from the virtual input samples (sentences) generated from the vicinity distribution of the source and target language sentences. With the augmented data, it performs simultaneous *self-learning* with an effective *distillation strategy* to learn a strongly adapted cross-lingual model from noisy (pseudo) labels for the target language task. We propose novel ways to generate virtual sentences using a multilingual masked LM (Conneau et al., 2020), and get reliable task labels by simultaneous multilingual co-training. This co-training employs a two-stage co-distillation process to ensure robust transfer to dissimilar and/or low-resource languages.

We validate the effectiveness and robustness of XLA by performing extensive experiments on three different zero-resource cross-lingual transfer tasks – XNER, XNLI, and PAWS-X, which posit different sets of challenges. We have experimented with many different language pairs (14 in total) comprising languages that are similar/dissimilar/low-resourced. XLA yields impressive results on XNER, setting SoTA in all tested languages outperforming the baselines by a good margin. In particular, the relative gains for XLA are higher for structurally dissimilar and/or low-resource languages, where the base model is weaker: 28.54%, 16.05%, and 9.25% absolute improvements for Urdu, Burmese, and Arabic, respectively. For XNLI, with only 5% labeled data in the source, it gets comparable results to the baseline that uses all the labeled data, and surpasses the standard baseline by 2.55% on average when it uses all the labeled data in the source. We also have similar findings in PAWS-X. We provide a comprehensive analysis of the factors that contribute to XLA's performance.

## 2 BACKGROUND

**Contextual representation and cross-lingual transfer**    In recent years, significant progress has been made in learning contextual word representations and pretrained models. Notably, BERT (Devlin et al., 2019) pretrains a Transformer (Vaswani et al., 2017) encoder with a masked language model (MLM) objective, and uses the same model architecture to adapt to a new task. It also comes with a multilingual version mBERT, which is trained jointly on 102 languages. RoBERTa (Liu et al., 2019) extends BERT with improved training, while XLM (Lample and Conneau, 2019) extends mBERT with a conditional LM and a translation LM (using parallel data) objectives. Conneau et al. (2020) train the largest multilingual language model XLM-R with RoBERTa framework.

Despite any explicit cross-lingual supervision, mBERT and its variants have been shown to learn cross-lingual representations that generalize well across languages. Wu and Dredze (2019) and Pires et al. (2019) evaluate the zero-shot cross-lingual transferability of mBERT on several tasks and attribute its generalization capability to shared subword units. Pires et al. (2019) also found structural similarity (*e.g.,* word order) to be another important factor for successful cross-lingual transfer. K et al. (2020), however, show that the shared subword has a minimal contribution; instead, the structural similarity between languages is more crucial for effective transfer (more in Appendix D).

**Vicinal risk minimization (VRM)**    Data augmentation supported by the VRM principle (Chapelle et al., 2001) can be an effective choice for achieving better out-of-distribution adaptation. In VRM, we minimize the empirical vicinal risk defined as: $\mathcal{L}_v(\theta) = \frac{1}{N} \sum_{n=1}^{N} l(f_\theta(\tilde{x}_n), \tilde{y}_n)$, where $f_\theta$ denotes the model parameterized by $\theta$, and $\tilde{\mathcal{D}} = \{(\tilde{x}_n, \tilde{y}_n)\}_{n=1}^{N}$ is an augmented dataset constructed by sampling the vicinal distribution $\vartheta(\tilde{x}_i, \tilde{y}_i | x_i, y_i)$ around the original training sample $(x_i, y_i)$. Defining vicinity is however quite challenging as it requires the extraction of samples from a distribution without hurting their labels. Earlier methods apply simple rules like rotation and scaling of images (Simard

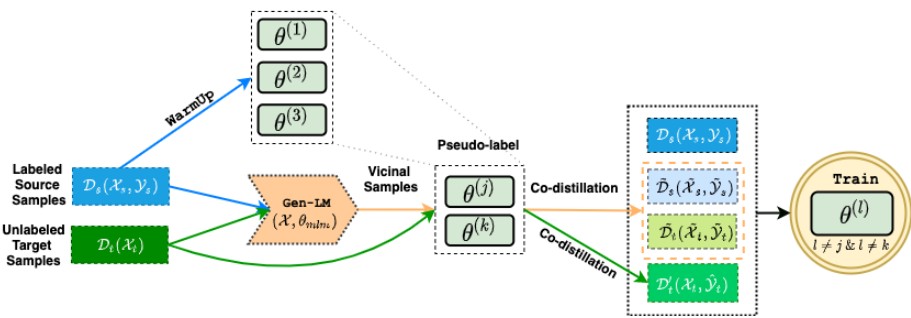

Figure 1: Training flow diagram of XLA. After training the base task models $\theta^{(1)}$, $\theta^{(2)}$, and $\theta^{(3)}$ on source labeled data $\mathcal{D}_s$ (**WarmUp**), we use two of them ($\theta^{(j)}$, $\theta^{(k)}$) to **pseudo-label** and **co-distill** the unlabeled target language data ($\mathcal{D}'_t$). A pretrained LM (**Gen-LM**) is used to generate new (vicinal) training samples for both source and target languages, which are also then pseudo-labeled and co-distilled using the two task models ($\theta^{(j)}$, $\theta^{(k)}$) to generate $\tilde{\mathcal{D}}_s$ and $\tilde{\mathcal{D}}_t$. The third model $\theta^{(l)}$ is then progressively trained on these datasets: $\{\mathcal{D}_s, \mathcal{D}'_t\}$ in epoch 1, $\tilde{\mathcal{D}}_t$ in epoch 2, and all in epoch 3.

et al., 1998). Recent work (Zhang et al., 2018; Berthelot et al., 2019) show impressive results in image classification with simple linear interpolation of data. However, to our knowledge, none of these methods have so far been successful in NLP due to the discrete nature of texts.[1]

**LM-based supervised augmentation**   Recently, a number of data-augmentation methods have been proposed using contextualized LMs like BERT, *e.g.,* Contextual Augmentation (Kobayashi, 2018), Conditional BERT (Wu et al., 2018), and AUG-BERT (Wu et al., 2018). These approaches use a constrained augmentation method which alters a pretrained LM to a label-conditional LM for a specific task. This means these methods update the parameters of the pretrained LM using the labels.

## 3   XLA FRAMEWORK

While recent cross-lingual transfer learning efforts have relied almost exclusively on multi-lingual pretraining and zero-shot transfer of a fine-tuned source model, there is a great potential for more elaborate methods that can leverage the unlabeled data better. Motivated by this, we present XLA - our unsupervised data augmentation framework for zero-resource cross-lingual task adaptation.

Figure 1 gives an overview of XLA. Let $\mathcal{D}_s = (\mathcal{X}_s, \mathcal{Y}_s)$ and $\mathcal{D}_t = (\mathcal{X}_t)$ denote the training data for a source language $s$ and a target language $t$, respectively. XLA augments data from various origins at different stages of training. In the initial stage (epoch 1), it uses the augmented training samples from the target language ($\mathcal{D}'_t$) along with the original source ($\mathcal{D}_s$). In later stages (epoch 2-3), it uses virtual (vicinal) sentences generated from the vicinity distribution of source and target examples: $\vartheta(\tilde{x}_n^s | x_n^s)$ and $\vartheta(\tilde{x}_n^t | x_n^t)$, where $x_n^s \sim \mathcal{X}_s$ and $x_n^t \sim \mathcal{X}_t$. It performs *self-training* on the augmented data to acquire the corresponding pseudo labels. To avoid *confirmation bias* with self-training where the model accumulates its own errors, it simultaneously trains three task models to generate *virtual* training data through data augmentation and filtering of potential label noises via multi-epoch *co-teaching* (Zhou and Li, 2005). In each epoch, the co-teaching process first performs *co-distillation*, where two peer task models are used to select "reliable" training examples to train the third model. The selected samples with pseudo labels are then added to the target task model's training data by taking the agreement from the other two models, a process we refer to as *co-guessing*. The co-distillation and co-guessing mechanism ensure robustness of XLA to out-of-domain distributions that can occur in a multilingual setup, *e.g.,* due to a structurally dissimilar and/or low-resource target language. Algorithm 1 gives a pseudocode of the overall training method.

Each of the *task* models in XLA is an instance of XLM-R fine-tuned on the source language task (e.g., English NER), whereas the pretrained masked LM parameterized by $\theta_{\mathrm{mlm}}$ (*i.e.,* before fine-tuning) is used to define the *vicinity* distribution $\vartheta(\tilde{x}_n | x_n, \theta_{\mathrm{mlm}})$ around each selected example $x_n$.

---

[1]Considering papers that have been published (or accepted) through peer review. There has been some concurrent work that uses pretrained LMs like BERT to craft *adversarial* examples (Li et al., 2020b). Although relevant, these methods have a different objective than ours, and none of them is cross- or multi-lingual.

---

**Algorithm 1** XLA: a robust unsupervised data augmentation framework for cross-lingual NLP

---

**Input:** *source* (s) and *target* (t) language datasets: $\mathcal{D}_s = (\mathcal{X}_s, \mathcal{Y}_s), \mathcal{D}_t = (\mathcal{X}_t)$; task models: $\theta^{(1)}, \theta^{(2)}, \theta^{(3)}$, pre-trained masked LM $\theta_{\mathrm{mlm}}$, mask ratio $P$, diversification factor $\delta$, sampling factor $\alpha$, and distillation factor $\eta$
**Output:** models trained on augmented data

1: $\theta^{(1)}, \theta^{(2)}, \theta^{(3)} = \text{WARMUP}(\mathcal{D}_s, \theta^{(1)}, \theta^{(2)}, \theta^{(3)})$           ▷ warm up with conf. penalty.
2: **for** $e \in [1:3]$ **do**                                   ▷ $e$ denotes epoch.
3:     **for** $k \in \{1, 2, 3\}$ **do**
4:        $\mathcal{X}_t^{(k)}, \mathcal{Y}_t^{(k)} = \text{DISTIL}(\mathcal{X}_t, \eta_e, \theta^{(k)})$        ▷ infer and select tgt training data for augmentation.
5:        **for** $j \in \{1, 2, 3\}$ **do**
6:           **if** $k == j$ **then** Continue
7:           /* source language data augmentation */
8:           $\tilde{\mathcal{X}}_s = \text{GEN-LM}(\mathcal{X}_s, \theta_{\mathrm{mlm}}, P, \delta)$          ▷ vicinal example generation.
9:           $\mathcal{X}_s^{(k)}, \mathcal{Y}_s^{(k)} = \text{DISTIL}(\tilde{\mathcal{X}}_s, \eta_e, \theta^{(k)}); \quad \mathcal{X}_s^{(j)}, \mathcal{Y}_s^{(j)} = \text{DISTIL}(\tilde{\mathcal{X}}_s, \eta_e, \theta^{(j)})$
10:           $\tilde{\mathcal{D}}_s = \text{AGREEMENT}\big(\mathcal{D}_s^{(k)} = (\mathcal{X}_s^{(k)}, \mathcal{Y}_s^{(k)}), \mathcal{D}_s^{(j)} = (\mathcal{X}_s^{(j)}, \mathcal{Y}_s^{(j)})\big)$
11:           /* target language data augmentation (no vicinity) */
12:           $\mathcal{X}_t^{(j)}, \mathcal{Y}_t^{(j)} = \text{DISTIL}(\mathcal{X}_t, \eta_e, \theta^{(j)})$
13:           $\mathcal{D}_t' = \text{AGREEMENT}\big(\mathcal{D}_t^{(k)} = (\mathcal{X}_t^{(k)}, \mathcal{Y}_t^{(k)}), \mathcal{D}_t^{(j)} = (\mathcal{X}_t^{(j)}, \mathcal{Y}_t^{(j)})\big)$      ▷ see line 4
14:           /* target language data augmentation */
15:           $\tilde{\mathcal{X}}_t = \text{GEN-LM}(\mathcal{X}_t, \theta\mathrm{mlm}, P, \delta)$         ▷ vicinal example generation.
16:           $\mathcal{X}_t^{(k)}, \mathcal{Y}_t^{(k)} = \text{DISTIL}(\tilde{\mathcal{X}}_t, \eta_e, \theta^{(k)}); \quad \mathcal{X}_t^{(j)}, \mathcal{Y}_t^{(j)} = \text{DISTIL}(\tilde{\mathcal{X}}_t, \eta_e, \theta^{(j)})$
17:           $\tilde{\mathcal{D}}_t = \text{AGREEMENT}\big(\mathcal{D}_t^{(k)} = (\mathcal{X}_t^{(k)}, \mathcal{Y}_t^{(k)}), \mathcal{D}_t^{(j)} = (\mathcal{X}_t^{(j)}, \mathcal{Y}_t^{(j)})\big)$
18:           /* train new models on augmented data */
19:           **for** $l \in \{1, 2, 3\}$ **do**
20:              **if** $l \neq j$ and $l \neq k$ **then**
21:                with sampling factor $\alpha$, train $\theta^{(l)}$ on $\mathcal{D}$,         ▷ train progressively
22:                where $\mathcal{D} = \{\mathcal{D}_s \mathbb{1}(e \in \{1, 3\}) \cup \mathcal{D}_t' \mathbb{1}(e \in \{1, 3\}) \cup \tilde{\mathcal{D}}_s \mathbb{1}(e = 3) \cup \tilde{\mathcal{D}}_t \mathbb{1}(e \in \{2, 3\})\}$
23: Return $\{\theta^{(1)}, \theta^{(2)}, \theta^{(3)}\}$

---

Although the data augmentation proposed in Contextual Augmentation (Kobayashi, 2018), Conditional BERT (Wu et al., 2018), AUG-BERT (Wu et al., 2018) also use a pretrained masked LM, there are some fundamental differences between our method and these approaches. Unlike these approaches our vicinity-based LM augmentation is purely unsupervised and we do not perform any fine-tuning of the pretrained vicinity model ($\theta_{lm}$). The vicinity model in XLA is a disjoint pretrained entity whose weights are not trained on any task objective. This disjoint characteristic gives our framework the flexibility to replace $\theta_{lm}$ even with a better monolingual LM for a specific target language, which in turn makes XLA extendable to utilize stronger LMs that may come in the future.

In the following, we describe the steps in Algorithm 1.

### 3.1 WARM-UP STEP: TRAINING TASK MODELS WITH CONFIDENCE PENALTY

We first train three instances of the XLM-R model ($\theta^{(1)}, \theta^{(2)}, \theta^{(3)}$) with an additional task-specific linear layer on the source language (English) labeled data. Each model has the same architecture (XLM-R large) but is initialized with different random seeds. For token-level prediction tasks (*e.g.,* NER), the *token-level* representations are fed into the classification layer, whereas for sentence-level tasks (*e.g.,* XNLI), the [CLS] representation is used as input to the classification layer.

**Training with confidence penalty** Our goal is to train the task models so that they can be used reliably for self-training on a target language that is potentially dissimilar and low-resourced. In such situations, an overly confident (overfitted) model may produce more noisy pseudo labels, and the noise will then accumulate as the training progresses. Overly confident predictions may also impose difficulties on our distillation methods (§3.3) in isolating good samples from noisy ones. However, maximum likelihood training with the standard cross-entropy (CE) loss may result in overfitted models that produce overly confident predictions (low entropy), especially when the class distribution is not balanced. We address this by adding a negative entropy term $-\mathcal{H}$ to the CE loss.

$$\mathcal{L}(\theta) = -\sum_{c=1}^{C} \Big[ \underbrace{y^c \log p_\theta^c(\mathbf{x})}_{\text{CE}} + \underbrace{p_\theta^c(\mathbf{x}) \log p_\theta^c(\mathbf{x})}_{-\mathcal{H}} \Big] \tag{1}$$

where $\mathbf{x}$ is the representation that goes to the output layer, and $y_n^c$ and $p_\theta^c(\mathbf{x}_n)$ are respectively the ground truth label and model predictions with respect to class $c$. Such regularizer of output distribution has been shown to be an effective generalization method for training large models (Pereyra et al., 2017). In our experiments (§4), we report significant gains with confidence penalty for cross-lingual transfer. Appendix C shows visualizations on why confidence penalty is helpful for distillation.

## 3.2 VICINITY DISTRIBUTION AND SENTENCE AUGMENTATION

Our augmentated sentences comes from two different sources: the *original* target language samples $\mathcal{X}_t$, and the *virtual* samples generated from the vicinity distribution of the source and target samples: $\vartheta(\tilde{x}_n^s | x_n^s, \theta_{\mathrm{mlm}})$ and $\vartheta(\tilde{x}_n^t | x_n^t, \theta_{\mathrm{mlm}})$, where $x_n^s \sim \mathcal{X}_s$ and $x_n^t \sim \mathcal{X}_t$. It has been shown that contextual LMs pretrained on large-scale datasets capture useful linguistic features and can be used to generate fluent grammatical texts (Hewitt and Manning, 2019). We use the XLM-R masked LM (Conneau et al., 2020) as our vicinity model $\theta_{\mathrm{mlm}}$, which is trained on massive multilingual corpora (2.5 TB of Common-Crawl data in 100 languages). Note that the vicinity model is a disjoint pretrained entity whose parameters are not trained on any task objective.

In order to generate samples around each *selected* example, we first randomly choose $P\%$ of the input tokens. Then we successively (*i.e.*, one at a time) mask one of the chosen tokens and ask $\theta_{\mathrm{mlm}}$ to predict a token in that masked position, *i.e.*, we compute $\vartheta(\tilde{x}_m | x, \theta_{\mathrm{mlm}})$ with $m$ being the index of the masked token. For a specific mask, we sample $S$ candidate words from the output distribution. We then generate novel sentences by following one of the two alternative approaches.

- **Successive max** In this approach, we take the most probable output token ($S = 1$) at each prediction step, $\hat{o}_m = \arg\max_o \vartheta(\tilde{x}_m = o | x, \theta_{\mathrm{mlm}})$. A new sentence is then constructed by $P\%$ newly generated tokens. We generate $\delta$ virtual samples for each original example $x$, by randomly masking $P\%$ tokens each time. Here, $\delta$ is the diversification factor.
- **Successive cross** In this approach, we divide each original sample $x$ into two parts and use successive max to create two sets of augmented samples of size $\delta_1$ and $\delta_2$ respectively. We then take the cross of these two sets to generate $\delta_1 \times \delta_2$ augmented samples.

Augmentation of sentences through *successive max* or *successive cross* is carried out within the GEN-LM (generate via LM) module in Algorithm 1. For tasks involving a single sequence (*e.g.*, XNER), we directly use successive max. Pairwise tasks like XNLI and PAWS-X have pairwise dependencies: dependencies between a premise and a hypothesis in XNLI or dependencies between a sentence and its possible paraphrase in PAWS-X. To model such dependencies, we use successive cross, which uses cross-product of two successive max applied independently to each component.

## 3.3 CO-LABELING OF AUGMENTED SENTENCES THROUGH CO-DISTILLATION

Traditional VRM based data augmentation methods assume that the samples generated by the vicinity model share the same class so that the same class labels can be used for the newly generated data (Chapelle et al., 2001). This approach does not consider the vicinity relation across examples of different classes. Recent methods relax this assumption and generate new images and their labels as simple *linear interpolations* (Berthelot et al., 2019). However, due to the discrete nature of texts, such linear interpolation methods have not been successful so far in NLP. The meaning of a sentence (*e.g.*, sentiment, word meanings) can change entirely even with minor variations in the original sentence. For example, consider the following example generated by our vicinity model (more in appendix G).

| | |
|---|---|
| **Original text:** | *EU rejects German call to boycott british lamb.* |
| **Masked text:** | *<mask> rejects german call to boycott british lamb.* |
| **MLM prediction:** | *Trump rejects german call to boycott british lamb.* |

Here, EU is an *Organization* whereas the newly predicted word *Trump* is a *Person* (different name type). Therefore, we need to relabel the augmented sentences no matter whether the original sentence has labels (source) or not (target). However, the relabeling process can induce noise, especially for dissimilar/low-resource languages, since the base task model may not be adapted fully in the early training stages. We propose a two-stage sample distillation process to filter out noisy augmented data.

**Sample distillation by single-model.** The first stage of distillation involves predictions from a single peer model for which we propose two alternatives:

(*i*) *Distillation by model confidence:* In this approach, we select samples based on the model's prediction confidence. This method is similar in spirit to the selection method proposed by Ruder and

Plank (2018). For sentence-level tasks (*e.g.,* XNLI), the model produces a single class distribution for each training example. In this case, the model's confidence is computed by $\hat{p} = \max_{c \in \{1...C\}} p_\theta^c(\mathbf{x})$. For token-level sequence labeling tasks (*e.g.,* NER), the model's confidence is computed by: $\hat{p} = \frac{1}{T}\big\{\max_{c \in \{1...C\}} p_\theta^c(\mathbf{x}_t)\big\}_{t=1}^{T}$, where $T$ is the length of the sequence. The distillation is then done by selecting the top $\eta\%$ samples with the highest confidence scores.

(*ii*) *Sample distillation by clustering:* We propose this method based on the finding that large neural models tend to learn good samples faster than noisy ones, leading to a lower loss for good samples and higher loss for noisy ones (Han et al., 2018; Arazo et al., 2019). We use a 1d two-component Gaussian Mixture Model (GMM) to model *per-sample loss distribution* and cluster the samples based on their *goodness*. GMMs provide flexibility in modeling the sharpness of a distribution and can be easily fit using *Expectation-Maximization* (EM) (Appendix B). The loss is computed based on the pseudo labels predicted by the model. For each sample $\mathbf{x}$, its *goodness* probability is the posterior probability $p(z = g|\mathbf{x}, \theta_{\mathrm{GMM}})$, where $g$ is the component with smaller mean loss. Here, distillation hyperparameter $\eta$ is the posterior probability threshold based on which samples are selected.

**Distillation by model agreement.** In the second stage of distillation, we select samples by taking the agreement (co-guess) of two different peer models $\theta^{(j)}$ and $\theta^{(k)}$ to train the third $\theta^{(l)}$. Formally,

$$\text{AGREEMENT}\big(\mathcal{D}^{(k)}, \mathcal{D}^{(j)}\big) = \{(\mathcal{X}^{(k)}, \mathcal{Y}^{(k)}) : \mathcal{Y}^{(k)} = \mathcal{Y}^{(j)}\} \quad s.t. \ k \neq j$$

### 3.4 DATA SAMPLES MANIPULATION

XLA uses multi-epoch co-teaching. It uses $\mathcal{D}_s$ and $\mathcal{D}'_t$ in the first epoch. In epoch 2, it uses $\tilde{\mathcal{D}}_t$ (target virtual), and finally it uses all the four datasets - $\mathcal{D}_s$, $\mathcal{D}'_t$, $\tilde{\mathcal{D}}_t$, and $\tilde{\mathcal{D}}_s$ (line 22 in Alg. 1). The datasets used at different stages can be of different sizes. For example, the number of augmented samples in $\tilde{\mathcal{D}}_s$ and $\tilde{\mathcal{D}}_t$ grow polynomially with the *successive cross* masking method. Also, the *co-distillation* produces sample sets of variable sizes. To ensure that our model does not overfit on one particular dataset, we employ a balanced sampling strategy. For $N$ number of datasets $\{\mathcal{D}_i\}_{i=1}^{N}$ with probabilities, $\{p_i\}_{i=1}^{N}$, we define the following multinomial distribution to sample from:

$$p_i = \frac{f_i^\alpha}{\sum_{j=1}^{N} f_j^\alpha} \text{ where } f_i = \frac{n_i}{\sum_{j=1}^{N} n_j} \tag{2}$$

where $\alpha$ is the sampling factor and $n_i$ is the total number of samples in the $i^{th}$ dataset. By tweaking $\alpha$, we can control how many samples a dataset can provide in the mix.

## 4 EXPERIMENTS

We consider three tasks in the *zero-resource* cross-lingual transfer setting. We assume labeled training data only in English, and transfer the trained model to a target language. For all experiments, we report the *mean score* of the three models that use different seeds (variance shown in Appendix F).

### 4.1 TASKS & SETTINGS

**XNER:** As a sequence labeling task, XNER evaluates the model's capability to learn task-specific contextual representations that depend on language structure. We use the standard CoNLL datasets (Sang, 2002; Sang and Meulder, 2003) for English (en), German (de), Spanish (es) and Dutch (nl). We also evaluate on Finnish (fi) and Arabic (ar) datasets collected from Bari et al. (2020). Note that Arabic is structurally different from English, and Finnish is from a different language family. To show how the models perform on extremely low-resource languages, we experiment with three structurally different languages from WikiANN (Pan et al., 2017) of different (unlabeled) training data sizes: Urdu (ur-20k training samples), Bengali (bn-10K samples), and Burmese (my-100 samples).

**XNLI** XNLI judges the model's ability to extract a reasonable meaning representation of sentences across different languages. We use the standard dataset (Conneau et al., 2018). For a given pair of sentences, the task is to predict the entailment relationship between the two sentences, *i.e.*, whether the second sentence (*hypothesis*) is an *Entailment*, *Contradiction*, or *Neutral* with respect to the first one (*premise*). We experiment with Spanish, German, Arabic, Swahili (sw), Hindi (hi) and Urdu.

**PAWS-X** The Paraphrase Adversaries from Word Scrambling Cross-lingual task (Yang et al., 2019a) requires the models to determine whether two sentences are paraphrases. We evaluate on all the six (typologically distinct) languages: fr, es, de, Chinese (zh), Japanese (ja), and Korean (ko).

**Settings** Our goal is to adapt a task model from a source (language) distribution to an unknown target (language) distribution assuming no labeled data in the target. In this scenario, there might be two different distributional gaps: (*i*) the generalization gap for the source distribution, and (*ii*) the gap between the source and target language distribution. We wish to investigate our method in tasks that exhibit such properties. We use the standard task setting for XNER, where we take 100% samples from the datasets as they come from various domains and sizes without any specific bias.

However, both XNLI and PAWS-X training data come with machine-translated texts in target languages. Thus, the data is parallel and lacks enough diversity (source and target come from the same domain). Cross-lingual models trained in this setup may pick up distributional bias (in the label space) from the source. Artetxe et al. (2020) also argue that the translation process can induce subtle artifacts that may have a notable impact on models. Therefore, for XNLI and PAWS-X, we experiment with two different setups. First, to ensure distributional differences and non-

Table 1: **F1 scores** in XNER on datasets from CoNLL and Bari et al. (2020). "−" represents no results were reported for the setup.

| Model | en | es | nl | de | ar | fi |
|---|---|---|---|---|---|---|
| **Supervised Results** (TRANSLATE-TRAIN-ALL) | | | | | | |
| LSTM-CRF (Bari et al., 2020) | 89.77 | 84.71 | 85.16 | 78.14 | 75.49 | 84.21 |
| XLM-R (Conneau et al., 2020) | 92.92 | 89.72 | 92.53 | 85.81 | − | − |
| XLM-R (our imp.) | 92.9 | 89.2 | 92.9 | 86.2 | 86.8 | 92.4 |
| **Zero-Resource Baseline** | | | | | | |
| mBERT$_{cased}$ (our imp.) | 91.13 | 74.76 | 79.58 | 70.99 | 45.48 | 65.95 |
| XLM-R (our imp.) | 92.23 | 79.29 | 80.87 | 73.40 | 49.04 | 75.57 |
| XLM-R (ensemble) | 92.76 | 80.62 | 81.46 | 75.40 | 52.30 | 76.85 |
| **Our Method** | | | | | | |
| mBERT$_{cased}$ +con-penalty | 90.81 | 75.06 | 79.26 | 72.31 | 47.03 | 66.72 |
| XLM-R+con-penalty | 92.49 | 80.45 | 81.07 | 73.76 | 49.94 | 76.05 |
| XLA | − | 83.05 | 85.21 | 80.33 | 57.35 | 79.75 |
| XLA (ensemble) | − | **83.24** | **85.32** | **80.99** | **58.29** | **79.87** |

parallelism, we use 5% of the training data from the source language and augment a different (nonparallel) 5% dataset for the target language. We used a different seed each time to retrieve the 5% target language data. Second, to compare with previous methods, we also evaluate on the standard 100% setup. However, the evaluation is done on the entire test set in both setups. We will refer to these two settings as **5%** and **100%**. Details about model settings and hyperparameters are in Appendix E.

## 4.2 RESULTS

**XNER** Table 1 reports the XNER results on the datasets from CoNLL and Bari et al. (2020), where we also evaluate an *ensemble* by averaging the probabilities from the three models. We observe that after performing *warm-up* with conf-penalty, XLM-R performs better than mBERT on average by ~3.8% for all the languages. On average, XLA gives a sizable improvement of ~5.5% on five different languages. Specifically, we get an absolute improvement of 3.76%, 4.34%, 6.94%, 8.31%, and 4.18% for *es, nl, de, ar,* and *fi*, respectively. Interestingly, XLA surpasses *supervised* LSTM-CRF for *nl* and *de* without using any target labeled data. It also produces comparable results for *es*.

In Table 2, we report the results on the three *low-resource* langauges from WikiANN. From these results and the results of *ar* and *fi* in Table 1, we see that XLA is very effective for languages that are structurally dissimilar and/or low-resourced, especially when the base model is weak: 28.54%, 16.05%, and 9.25% absolute improvements for ur, my and ar, respectively.

**XNLI-5%** From Table 3, we see that the performance of XLM-R trained on 5% data is surprisingly good compared to the model trained on full data (XLM-R (our imp.)), lagging by only 5.6% on average. In our single GPU implementation of XNLI, we could not reproduce the reported results of Conneau et al. (2020). However, our results resemble the reported XLM-R results of XTREME (Hu et al., 2020). We consider XTREME as our standard baseline for XNLI-100%.

Table 2: XNER results on WikiANN

| Model | ur | bn | my |
|---|---|---|---|
| **Supervised Results** | | | |
| XLM-R (our-impl) | 97.1 | 97.8 | 76.8 |
| **Zero-Resource Results** | | | |
| XLM-R (XTREME) | 56.4 | 78.8 | 54.3 |
| XLM-R (our imp.) | 56.45 | 78.17 | 54.56 |
| XLA | **84.99** | **82.68** | **70.61** |

We observe that with only 5% labeled data in the source, XLA gets comparable results to the XTREME baseline that uses 100% labeled data (lagging behind by only ~0.7% on average); even for *ar* and *sw*, we get 0.22% and 1.11% improvements, respectively. It surpasses the standard 5% baseline by 4.2% on average. Specifically, XLA gets absolute improvements of 3.05%, 3.34%, 5.38%, 5.01%, 4.29%, and 4.12% for *es, de, ar, sw, hi,* and *ur*, respectively. Again, the gains are relatively higher for low-resource and/or dissimilar languages despite the base model being weak in such cases.

**XNLI-100%** Now, considering XLA's performance on the full (100%) labeled source data in Table 3, we see that it achieves state-of-the-art results for all of the languages with an absolute improvement of 2.55% on average from the XTREME baseline. Specifically, XLA gets absolute improvements of 1.95%, 1.68%, 4.30%, 3.50%, 3.24%, and 1.65% for *es, de, ar, sw, hi,* and *ur*, respectively.

**PAWS-X** Similar to XNLI, we observe sizable improvements for XLA over the baselines on PAWS-X for both 5% and 100% settings (Table 4). Specifically, in 5% setting, XLA gets absolute gains of 5.33%, 5.94%, 5.04%, 6.85%, 7.00%, and 5.45% for *de, es, fr, ja, ko,* and *zh*, respectively, while in 100% setting, it gets 2.21%, 2.36%, 2.00%, 3.99%, 4.53%, and 4.41% improvements respectively. In general, we get an average improvements of 5.94% and 3.25% in PAWS-X-5% and PAWS-X-100% settings respectively. Moreover, our 5% setting outperforms 100% XLM-R baselines for *es, ja,* and *zh*. Interestingly, in the 100% setup, our XLA (ensemble) achieves almost similar accuracies compared to supervised finetuning of XLM-R on all target language training dataset.

Table 3: Results in accuracy for XNLI.

| Model | en | es | de | ar | sw | hi | ur |
|---|---|---|---|---|---|---|---|
| **Supervised Results** (TRANSLATE-TRAIN-ALL) | | | | | | | |
| XLM-R | 89.1 | 86.6 | 85.7 | 83.1 | 78.0 | 81.6 | 78.1 |
| Zero-Resource Baseline for **Full (100%) English labeled** training set | | | | | | | |
| XLM-R (XTREME) | 88.7 | 83.7 | 82.5 | 77.2 | 71.2 | 75.6 | 71.7 |
| XLM-R (our imp.) | 88.87 | 84.34 | 82.78 | 78.44 | 72.08 | 76.40 | 72.10 |
| XLM-R (ensemble) | 89.24 | 84.73 | 83.27 | 79.06 | 73.17 | 77.23 | 73.07 |
| XLM-R+con-penalty | 88.83 | 84.30 | 82.86 | 78.20 | 71.83 | 76.24 | 71.62 |
| XLA | – | 85.65 | 84.15 | 80.50 | 74.70 | 78.74 | 73.35 |
| XLA (ensemble) | – | **86.12** | **84.61** | **80.89** | **74.89** | **78.98** | **73.45** |
| Zero-Resource Baseline for **5% English labeled** training set | | | | | | | |
| XLM-R (our imp.) | 83.08 | 78.48 | 77.54 | 72.04 | 67.3 | 70.41 | 66.72 |
| XLM-R (ensemble) | 84.65 | 79.56 | 78.38 | 72.22 | 66.93 | 71.00 | 66.79 |
| XLM-R+con-penalty | 84.24 | 79.23 | 78.47 | 72.43 | 67.72 | 71.08 | 67.63 |
| XLA | – | 81.53 | 80.88 | 77.42 | 72.31 | 74.70 | 70.84 |
| XLA (ensemble) | – | **82.35** | **81.93** | **78.56** | **73.53** | **75.20** | **71.15** |

## 5 ANALYSIS

In this section, we further analyze XLA by dissecting it and measuring the contribution of its different components. For this, we use the XNER task and analyze the model based on the results in Table 1.

### 5.1 ANALYSIS
#### OF DISTILLATION METHODS

**Model confidence vs. clustering** We first analyze the performance of our *single-model distillation* methods (§3.3) to see which of the two alternatives works better. From Table 5, we see that both perform similarly, with *model confidence* being slightly better. In our main experiments (Tables 1-4) and subsequent analysis, we use model confidence for distillation. However, we should not rule out the clustering method as it gives a more general solution to consider other distillation features (*e.g.,* sequence length, language) than model prediction scores, which we did not explore in this paper.

Table 4: Results in accuracy for PAWS-X.

| Model | en | de | es | fr | ja | ko | zh |
|---|---|---|---|---|---|---|---|
| **Supervised Results** (TRANSLATE-TRAIN-ALL) | | | | | | | |
| XLM-R (our impl.) | 95.8 | 92.5 | 92.8 | 93.5 | 85.5 | 86.6 | 87.6 |
| Zero-Resource Baseline for **Full (100%) English labeled** training set | | | | | | | |
| XLM-R (XTREME) | 94.7 | 89.7 | 90.1 | 90.4 | 78.7 | 79.0 | 82.3 |
| XLM-R (our imp.) | 95.46 | 90.06 | 89.92 | 90.85 | 79.89 | 79.74 | 82.49 |
| XLM-R (ensemble) | 96.10 | 90.75 | 90.55 | 91.80 | 80.55 | 80.70 | 83.45 |
| XLM-R+con-penalty | 95.38 | 90.75 | 90.72 | 91.71 | 81.77 | 82.07 | 84.25 |
| XLA | – | 92.27 | 92.28 | 92.85 | 83.88 | 84.27 | 86.90 |
| XLA (ensemble) | – | **92.55** | **92.35** | **93.35** | **84.30** | **84.35** | **86.95** |
| Zero-Resource Baseline for **5% English labeled** training set | | | | | | | |
| XLM-R (our imp.) | 91.15 | 83.72 | 84.32 | 85.08 | 73.65 | 72.60 | 77.22 |
| XLM-R (ensemble) | 92.05 | 84.05 | 84.65 | 85.75 | 74.30 | 71.95 | 77.50 |
| XLM-R+con-penalty | 91.85 | 86.15 | 86.38 | 85.98 | 76.03 | 75.43 | 79.15 |
| XLA | – | 89.05 | 90.27 | 90.12 | 80.50 | 79.60 | 82.65 |
| XLA (ensemble) | – | **89.25** | **90.85** | **90.25** | **81.15** | **80.15** | **82.90** |

**Distillation factor** $\eta$   We next show the results for different distillation factor ($\eta$) in Table 5. Here 100% refers to the case when no single-model distillation is done based on model confidence. We notice that the best results for each of the languages are obtained for values other than 100%, which indicates that distillation is indeed an effective step in XLA. See Appendix C.2 for more on $\eta$.

**Two-stage distillation** We now validate whether the second-stage distillation (*distillation by model agreement*) is needed. In Table 5, we also compare the results with the model agreement (shown as $\cap$) to the results without using any agreement (shown as $\phi$). We observe better performance with *model agreement* in all the cases on top of the single-model distillation, which validates its utility.

### 5.2 DIFFERENT TYPES OF AUGMENTATION IN DIFFERENT STAGES

Figure 2 presents the effect of different types of augmented data used by different epochs in our multi-epoch co-teaching framework. We observe that in every epoch, there is a significant boost in

Table 5: Analysis of **distillation** on XNER. Results after epoch-1 training that uses $\{\mathcal{D}_s, \mathcal{D}'_t\}$.

| $\eta$ | Agreement | es | nl | de | ar | fi |
|---|---|---|---|---|---|---|
| | | Distillation by **clustering** | | | | |
| 0.7 | $\cap$ | 82.28 | 83.25 | 78.86 | 52.64 | 78.47 |
| 0.5 | $\cap$ | 82.35 | 83.11 | 78.16 | 54.20 | 78.28 |
| | | Distillation by **model confidence** | | | | |
| 50% | $\cap$ | **82.52** | 82.46 | 75.95 | 52.00 | 77.51 |
| | $\phi$ | 81.66 | 82.26 | 77.19 | 52.97 | 77.77 |
| 80% | $\cap$ | 82.33 | **83.53** | 78.50 | **54.48** | 78.43 |
| | $\phi$ | 81.61 | 83.03 | 77.08 | 53.31 | 78.34 |
| 90% | $\cap$ | 81.90 | 82.80 | **79.03** | 52.41 | **78.66** |
| | $\phi$ | 81.21 | 82.77 | 77.28 | 52.20 | 77.93 |
| 100% | $\cap$ | 82.50 | 82.35 | 77.06 | 52.58 | 77.51 |
| | $\phi$ | 81.89 | 82.15 | 76.97 | 52.68 | 78.01 |

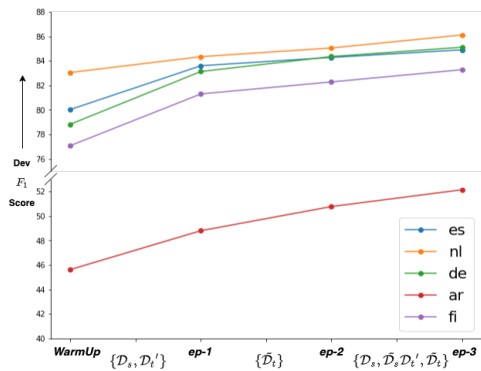

Figure 2: Validation F1 results in XNER for multi-epoch co-teaching training of XLA.

F1 scores for each of the languages. Arabic, being structural dissimilar to English, has a lower base score, but the relative improvements brought by XLA are higher for Arabic, especially in epoch 2 when it gets exposed to the target language virtual data ($\tilde{\mathcal{D}}_t$) generated by the vicinity distribution.

## 5.3 ROBUSTNESS OF XLA

Table 6 shows the robustness of the finetuned XLA model on XNER task. After finetuning in a specific target language, the F1 scores in English remain almost similar. For some languages, XLA adaptation on a different language also improves the performance. For example, Arabic gets improvements for all XLA-adapted models (compare 50.88 with others). This indicates that augmentation of XLA does not overfit on a target language.

Table 6: **F1 scores** on XNER for all target languages. Each column (*e.g.,* es) under **XLA Model** represents results in all target languages for a XLA model trained with the augmented data in a specific language (*e.g.,* es). The **Zero shot+con-penalty** column represents the zero-shot results for the model after **WarmUp**.

| Tgt lang | Zero shot + con-penalty | XLA Model | | | | |
|---|---|---|---|---|---|---|
| | | es | nl | de | ar | fi |
| en | **92.88** | 92.92 | 92.87 | 92.91 | 92.80 | 92.68 |
| es | 81.42 | **83.24** | 82.01 | 77.71 | 80.29 | 81.97 |
| nl | 81.27 | 81.22 | **85.32** | 80.54 | 82.36 | 84.20 |
| de | 75.20 | 73.63 | 75.03 | **80.03** | 76.97 | 73.77 |
| ar | 50.88 | 52.66 | 53.08 | 52.52 | **58.29** | 53.80 |
| fi | 76.97 | 77.02 | 77.06 | 76.69 | 77.13 | **80.11** |

## 5.4 EFFECT
## OF CONFIDENCE PENALTY & ENSEMBLE

For all the three tasks, we get reasonable improvements over the baselines by training with confidence penalty regularizer (§3.1). Specifically, we get 0.56%, 0.74%, 1.89%, and 1.18% improvements in XNER, XNLI-5%, PAWS-X-5%, and PAWS-X-100% respectively (Table 1,3,4). The improvements in XNLI-100% are marginal and inconsistent, which we suspect due to the balanced class distribution.

From the results of ensemble models, we see that the ensemble boosts the baseline XLM-R. However, our regular XLA still outperforms the ensemble baselines by a sizeable margin. Moreover, ensembling the trained models from XLA further improves the performance. These comparisons ensure that the capability of XLA through co-teaching and co-distillation is beyond the ensemble effect.

## 6 CONCLUSION

We propose a novel data augmentation framework, XLA, for zero-resource cross-lingual task adaptation. XLA performs simultaneous self-training with data augmentation and unsupervised sample selection. With extensive experiments on three different cross-lingual tasks spanning many language pairs, we have demonstrated the effectiveness of XLA. For the zero-resource XNER task, XLA sets a new SoTA for all the tested languages. For both XNLI and PAWS-X tasks, with only 5% labeled data in the source, XLA gets comparable results to the baseline that uses 100% labeled data. Through an in-depth analysis, we show the cumulative contributions of different components of XLA.

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

APPENDIX

Here we provide the additional contents regarding the XLA framework. In appendix A, we present the justification for design choices of XLA framework. In appendix B, we discuss the mathematical details of EM training for two components GMM clustering algorithm. In appendix C, we visualize various effects of confidence penalty. In appendix E, we present the setup details of our experiments. In appendix D and F, we elaborate on the related work and results with standard deviation as well as comparing with prior research work, respectively. Finally, in appendix G we present the examples of augmented samples generated by our vicinity model for XNER, XNLI, and PAWS-X.

## A    JUSTIFICATIONS FOR DESIGN METHODOLOGY OF XLA FRAMEWORK

Here are our justifications for various design principles of the XLA framework.

**Is using three models with different initialization necessary?**    Yes, different initialization ensures different convergence paths, which results in diversity during inference. Co-labeling (Section 3.3) utilizes this property. There could be some other ways to achieve the same thing. For example, our initial attempt with three different heads (sharing a backbone net) didn't work well.

**Is using three epochs necessary?**    We utilize different types of datasets in different epochs. While pseudo-labeling may induce noise, the model's predictions for in-domain cross-lingual samples are usually better. Because of this, for a smooth transition, we apply the vicinal samples in the second epoch. Finally, inspired by the joint training of the cross-lingual language model, in the third epoch we use all four datasets. We also include the labeled source data which ensures that our model does not overfit on target distribution as well as persists the generalization capability of the source distribution.

**Need for the combination of co-teaching, co-distillation and co-guessing?**    The combination of these helps to distill out the noisy samples better.

**Efficiency of the method and expensive extra costs for large-scale pretrained models**    It is a common practice in model selection to train 3-5 disjoint LM-based task models (e.g., XLM-R on NER) with different random seeds and report the ensemble score or score of the best (validation set) model. In contrast, XLA uses 3 different models and jointly trains them where the models assist each other through distillation and co-labeling. In that sense, the extra cost comes from distillation and co-labeling, which is not significant and is compensated by the significant improvements that XLA offers.

## B    DETAILS ON DISTILLATION BY CLUSTERING

One limitation of the confidence-based (single-model) distillation is that it does not consider task-specific information. Apart from classifier confidence, there could be other important features that can distinguish a good sample from a noisy one. For example, for sequence labeling, *sequence length* can be an important feature as the models tend to make more mistakes (hence noisy) for longer sequences Bari et al. (2020). One might also want to consider other features like *fluency*, which can be estimated by a pre-trained conditional LM like GPT Radford et al. (2019). In the following, we introduce a clustering-based method that can consider these additional features to separate good samples from bad ones.

Here our goal is to cluster the samples based on their *goodness*. It has been shown in computer vision that deep models tend to learn good samples faster than noisy ones, leading to a lower loss for good samples and higher loss for noisy ones Han et al. (2018); Arpit et al. (2017). We propose to model *per-sample loss distribution* (along with other task-specific features) with a mixture model, which we fit using an *Expectation-Maximization* (EM) algorithm. However, contrary to those approaches which use actual (supervised) labels, we use the model predicted pseudo labels to compute the loss for the samples.

We use a two-component Gaussian Mixture Model (GMM) due to its flexibility in modeling the sharpness of a distribution Li et al. (2020a). In the following, we describe the EM training of the

GMM for one feature, *i.e.,* per-sample loss, but it is trivial to extend it to consider other indicative task-specific features like sequence length or fluency score (see any textbook on machine learning).

**EM training for two-component GMM** Let $x_i \in \mathbb{R}$ denote the loss for sample $\mathbf{x}_i$ and $z_i \in \{0, 1\}$ denote its cluster id. We can write the 1d GMM model as:

$$p(x_i|\theta, \pi) = \sum_{k=0}^{1} \mathcal{N}(x_i|\mu_k, \sigma_k)\pi_k \tag{3}$$

where $\theta_k = \{\mu_k, \sigma_k^2\}$ are the parameters of the $k$-th mixture component and $\pi_k = p(z_i = k)$ is the probability (weight) of the $k$-th component with the condition $0 \leq \pi_k \leq 1$ and $\sum_k \pi_k = 1$.

In EM, we optimize the *expected complete data* log likelihood $Q(\theta, \theta^{t-1})$ defined as:

$$
\begin{align}
Q(\theta, \theta^{t-1}) &= \mathbb{E}(\sum_i \log[p(x_i, z_i|\theta)]) \tag{4} \\
&= \mathbb{E}(\sum_i \sum_k \mathbb{I}(z_i = k)\log[p(x_i|\theta_k)\pi_k]) \tag{5} \\
&= \sum_i \sum_k \mathbb{E}(\mathbb{I}(z_i = k))\log[p(x_i|\theta_k)\pi_k] \tag{6} \\
&= \sum_i \sum_k p(z_i = k|x_i, \theta^{t-1})\log[p(x_i|\theta_k)\pi_k] \tag{7} \\
&= \sum_i \sum_k r_{i,k}(\theta^{t-1})\log p(x_i|\theta_k) + r_{i,k}(\theta^{t-1})\log \pi_k \tag{8}
\end{align}
$$

where $r_{i,k}(\theta^{t-1})$ is the responsibility that cluster $k$ takes for sample $\mathbf{x}_i$, which is computed in the E-step so that we can optimize $Q(\theta, \theta^{t-1})$ (Eq. 8) in the M-step. The E-step and M-step for a 1d GMM can be written as:

**E-step:** Compute $r_{i,k}(\theta^{t-1}) = \frac{\mathcal{N}(x_i|\theta_k^{t-1})\pi_k^{t-1}}{\sum_k \mathcal{N}(x_i|\theta_k^{t-1})\pi_k^{t-1}}$

**M-step:** Optimize $Q(\theta, \theta^{t-1})$ *w.r.t.* $\theta$ and $\pi$

- $\pi_k = \frac{\sum_i r_{i,k}}{\sum_i \sum_k r_{i,k}} = \frac{1}{N}\sum_i r_{i,k}$
- $\mu_k = \frac{\sum_i r_{i,k}x_i}{\sum_i r_{i,k}}; \qquad \sigma_k^2 = \frac{\sum_i r_{i,k}(x_i - \mu_k)^2}{\sum_i r_{i,k}}$

**Inference** For a sample $\mathbf{x}$, its *goodness* probability is the posterior probability $p(z = g|\mathbf{x}, \theta)$, where $g \in \{0, 1\}$ is the component with smaller mean loss. Here, distillation hyperparameter $\eta$ is the posterior probability threshold based on which samples are selected.

**Relation with *distillation by model confidence*** Astute readers might have already noticed that per-sample loss has a direct deterministic relation with the model confidence. Even though they are different, these two distillation methods consider the same source of information. However, as mentioned, the clustering-based method allows us to incorporate other indicative features like length, fluency, etc. For a fair comparison between the two methods, we use only the per-sample loss in our primary (single-model) distillation methods.

## C    VISUALIZING THE EFFECT OF CONFIDENCE PENALTY

### C.1    EFFECT OF CONFIDENCE PENALTY IN CLASSIFICATION

In Figure 3, we present the effect of the confidence penalty (Eq. 1 in the main paper) in the target language (*Spanish*) classification on the XNER dev. data (*i.e.,* after training on English NER). We show the class distribution from the final logits (on the target language) using t-SNE plots van der Maaten and Hinton (2008).

From the figure, it is evident that the use of confidence penalty in the warm-up step makes the model more robust to unseen out-of-distribution target language data yielding better predictions, which in turn also provides a better *prior* for self-training with pseudo labels.

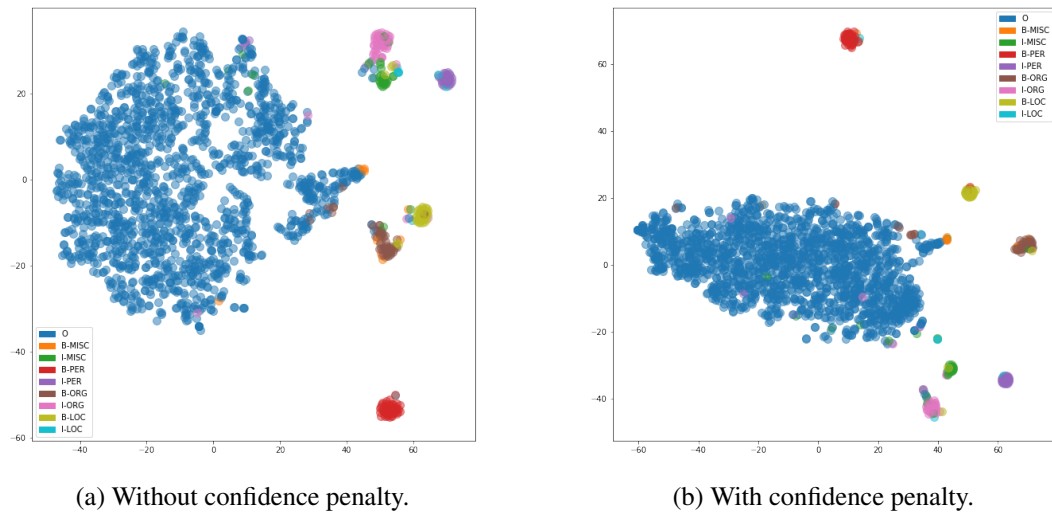

(a) Without confidence penalty.         (b) With confidence penalty.

Figure 3: Effect of training with confidence penalty in the warm-up step on target (*Spanish*) language XNER classification using t-SNE plots. From the visualization, it can be seen that the model trained with confidence penalty shows better inter-class separation which exhibits robustness of the multilingual model.

## C.2 EFFECT OF CONFIDENCE PENALTY IN LOSS DISTRIBUTION

Figures 4(a) and 4(b) present the per-sample loss (*i.e.*, mean loss per sentence *w.r.t.* the pseudo labels) distribution in histogram without and with confidence penalty, respectively. Here, *accurate-2* refers to the sentences which have at most two wrong NER labels, and sentences containing more than two errors are referred to as *noisy* samples. It shows that without confidence penalty, there are many noisy samples with a small loss which is not desired. In addition to that, the figures also suggest that the confidence penalty helps to separate the clean samples from the noisy ones either by clustering or by model confidence.

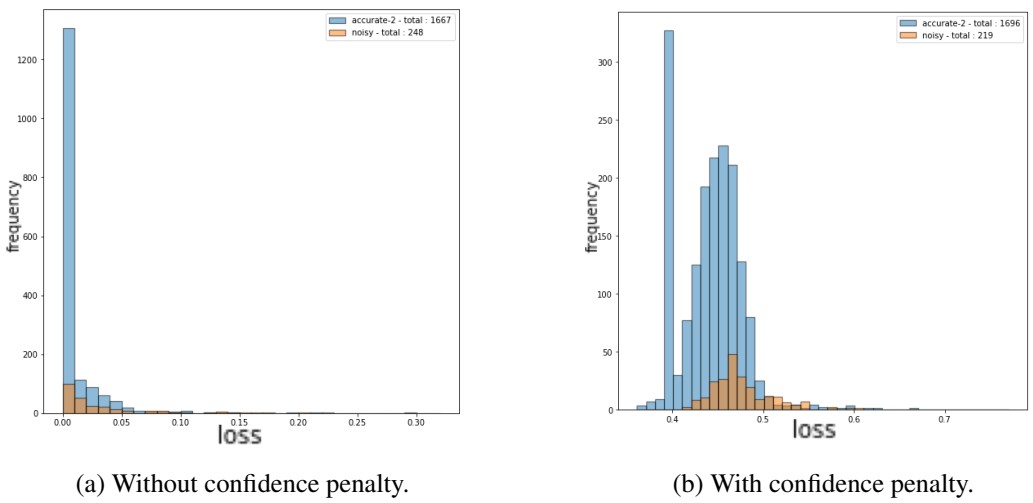

(a) Without confidence penalty.         (b) With confidence penalty.

Figure 4: Histogram of loss distribution on target (*Spanish*) language XNER classification.

Figures 5(a) and 5(b) present the loss distribution in a scatter plot by sorting the sentences based on their length in the x-axis; y-axis represents the loss. As we can see, the losses are indeed more scattered when we train the model with confidence penalty, which indicates higher per-sample entropy, as expected. Also, we can see that as the sentence length increases, there are more wrong predictions. Our distillation method should be able to distill out these noisy pseudo samples.

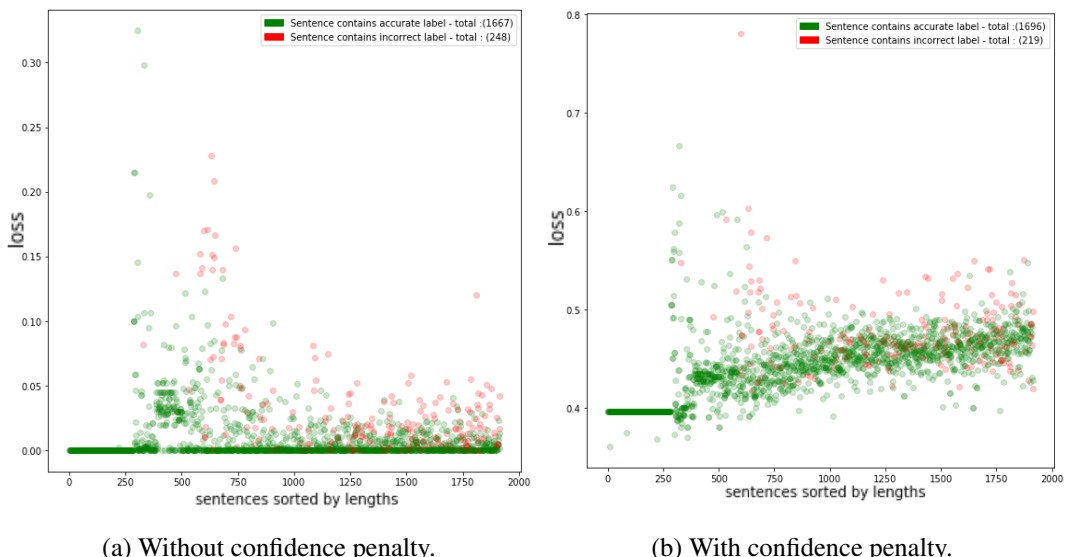

(a) Without confidence penalty.          (b) With confidence penalty.

Figure 5: Scatter plot of loss distribution on target (*Spanish*) language XNER classification.

Finally, Figures 6(a) and 6(b) show the length distribution of all vs. the selected sentences (by *Distillation by model confidence*) without and with confidence penalty. Bari et al. (2020) shows that cross-lingual NER inference is heavily dependent on the length distribution of the samples. In general, the performance of the lower length samples is more accurate. However, if we only select the lower length samples we will easily overfit. From these plots, we observe that the confidence penalty also helps to perform a better distillation as more sentences are selected (by the distillation procedure) from the lower length distribution, while still covering the entire lengths. This shows that using the confidence penalty in training, model becomes more robust.

In summary, comparing the Figures 4 - 6, we can conclude that training without confidence penalty can make the model more prone to over-fitting, resulting in more noisy pseudo labels. Training with confidence penalty not only improves pseudo labeling accuracy but also helps the distillation methods to perform better noise filtering.

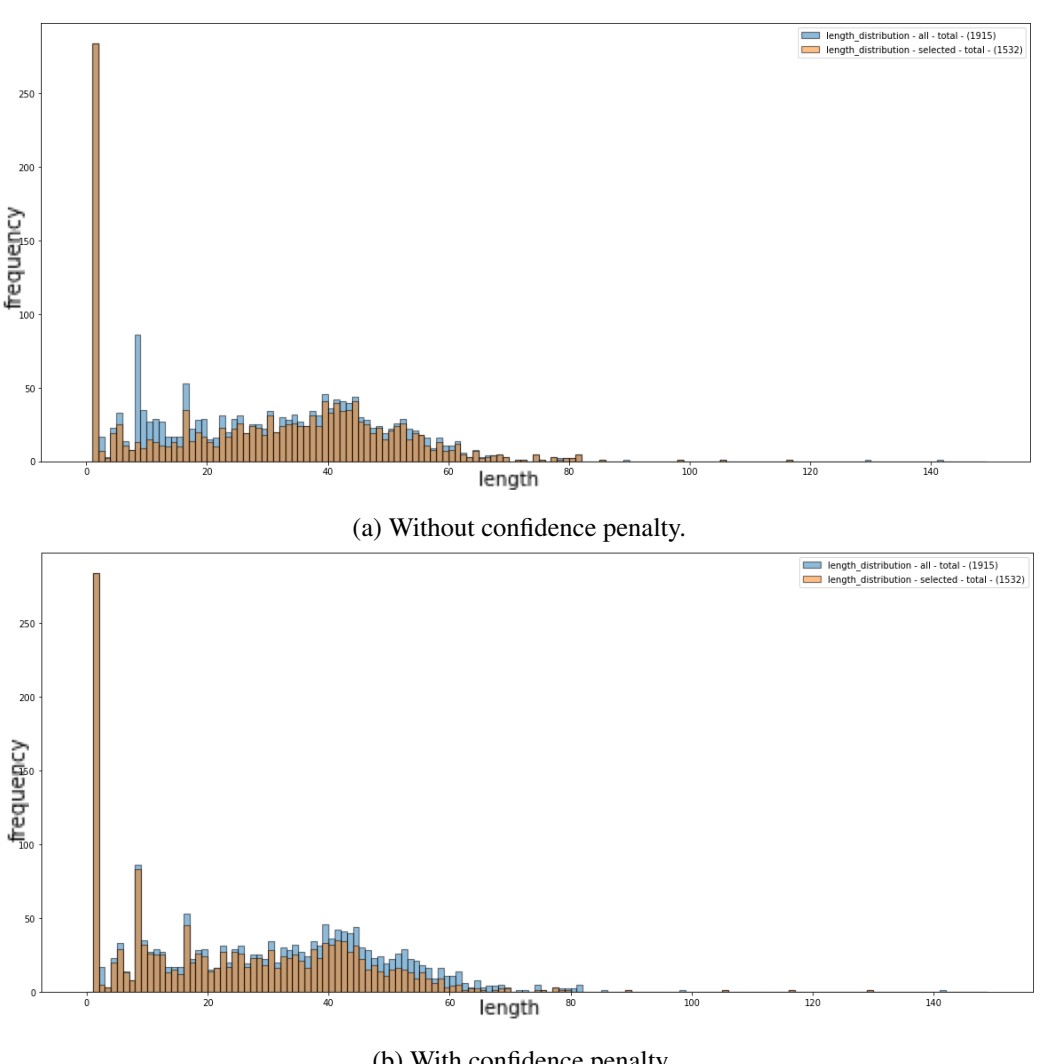

(a) Without confidence penalty.

(b) With confidence penalty.

Figure 6: Distribution of selected sentence lengths on target (*Spanish*) language XNER classification.

# D  EXTENDED RELATED WORK

**Contextual representation and cross-lingual transfer.**  In earlier approaches, word representations are learned from simple variants of the skip-gram model Mikolov et al. (2013), where each word has a single representation regardless of its context Grave et al. (2018); Pennington et al. (2014). Recent approaches learn word representations that change based on the context that the word appears in McCann et al. (2017); Peters et al. (2018); Howard and Ruder (2018); Devlin et al. (2019); Yang et al. (2019b); Radford et al. (2019).

Peters et al. (2018) propose ELMo - a bidirectional LSTM-based LM pre-training method for learning contextualized word representations. ELMo uses a linear combination of all of its layers' representations for predicting on a target task. However, because of sequential encoding, LSTM-based LM pre-training is hard to train at scale. Vaswani et al. (2017) propose the Transformer architecture based on multi-headed self-attention and positional encoding. The Transformer encoder can capture long-range sequential information and allows constant time encoding of a sequence through parallelization. Radford et al. (2018) propose GPT-1, which pre-trains a Transformer decoder with a conditional language model objective and then fine-tune it on the task with minimal changes to the model architecture. In the same spirit, Devlin et al. (2019) propose BERT, which pre-trains a Transformer encoder with a masked language model (MLM) objective, and uses the same model architecture to adapt to a new task. The advantage of MLM objective is that it allows bidirectional encoding, whereas the standard (conditional) LM is unidirectional (*i.e.*, uses either left context or right context).

BERT also comes with a multilingual version called mBERT, which has 12 layers, 12 heads and 768 hidden dimensions, and it is trained jointly on 102 languages with a shared vocabulary of 110K subword tokens.[2] Despite any explicit cross-lingual supervision, mBERT has been shown to learn cross-lingual representations that generalise well across languages. Wu and Dredze (2019); Pires et al. (2019) evaluate the zero-shot cross-lingual transferability of mBERT on several NLP tasks and attribute its generalization capability to shared subword units. Pires et al. (2019) additionally found structural similarity (*e.g.*, word order) to be another important factor for successful cross-lingual transfer. K et al. (2020), however, show that the shared subword has minimal contribution, rather the structural similarity between languages is more crucial for effective transfer. Artetxe et al. (2019) further show that joint training may not be necessary and propose an alternative method to transfer a monolingual model to a bi-lingual model through learning only the word embeddings in the target language. They also identify the vocabulary size per language as an important factor.

Lample and Conneau (2019) extend mBERT with a conditional LM and a translation LM (using parallel data) objectives and a language embedding layer. They train a larger model with more monolingual data. Huang et al. (2019) propose to use auxiliary tasks such as cross-lingual word recovery and paraphrase detection for pre-training. Recently, Conneau et al. (2020) train the largest multilingual language model with 24-layer transformer encoder, 1024 hidden dimensions and 550M parameters. Keung et al. (2019) use adversarial fine-tuning of mBERT to achieve better language invariant contextual representation for cross-lingual NER and MLDoc document classification.

**Vicinal risk minimization.**  One of the fundamental challenges in deep learning is to train models that generalize well to examples outside the training distribution. The widely used Empirical Risk Minimization (ERM) principle where models are trained to minimize the average training error has been shown to be insufficient to achieve generalization on distributions that differ slightly from the training data Szegedy et al. (2014); Zhang et al. (2018). Data augmentation supported by the Vicinal Risk Minimization (VRM) principle Chapelle et al. (2001) can be an effective choice for achieving better out-of-training generalization.

In VRM, we minimize the empirical vicinal risk defined as:

$$\mathcal{L}_v(\theta) = \frac{1}{N} \sum_{n=1}^{N} l(f_\theta(\tilde{x}_n), \tilde{y}_n) \tag{9}$$

where $f_\theta$ denotes the model parameterized by $\theta$, and $\mathcal{D}^{\text{aug}} = \{(\tilde{x}_n, \tilde{y}_n)\}_{n=1}^{N}$ is an augmented dataset constructed by sampling the vicinal distribution $\vartheta(\tilde{x}, \tilde{y}|x_i, y_i)$ around the original training sample

---

[2]github.com/google-research/bert/blob/master/multilingual.md

$(x_i, y_i)$. Defining vicinity is however challenging as it requires to extract samples from a distribution without hurting the labels. Earlier methods apply simple rules like rotation and scaling of images Simard et al. (1998). Recently, Zhang et al. (2018); Berthelot et al. (2019) and Li et al. (2020a) show impressive results in image classification with simple linear interpolation of data. However, to our knowledge, none of these methods has so far been successful in NLP due to the discrete nature of texts.

# E  SETUP DETAILS

## E.1  ZERO-SHOT VS. ZERO-RESOURCE TRANSFER

Previous work on cross-lingual transfer has followed different training-validation standards. Xie et al. (2018) perform cross-lingual transfer of NER from a source language to a target language, where they train their model on translations of the source language training data and validate it (for model selection) with target language development data. They call this as an *unsupervised setup* as they use an unsupervised word translation model Conneau et al. (2017). Several other studies Conneau et al. (2018); Lample and Conneau (2019); Wang et al. (2019) also apply the same setting and select their model based on target language development set performance. On the other hand, Artetxe and Schwenk (2018), Wu and Dredze (2019) validate their models using source language development data. Bari et al. (2020) show significant performance differences between validation with source vs. target language development data for NER. Later, Conneau et al. (2020) provide a comprehensive analysis of different training-validation setups and encourage validating with the source language development data. Therefore, it is clear that there is no unanimous agreement regarding the proper setup. Following the previous work and landscape of the problem, we think that different settings should be considered under different circumstances.

In a pure *zero-shot* cross-lingual transfer, no target language data should be used either for training or for model selection. The goal here is to evaluate the *generalizability* and *transferability* of a model trained on a known source language distribution to an unknown target language distribution. In this sense, zero-shot setting is suitable to measure the cross-lingual transferability of a pre-trained model.

Our goal in this work is not to propose a new pre-training approach, rather to propose novel cross-lingual adaptation methods and evaluate their capability on downstream tasks. Our proposed XLA framework performs simultaneous self-training with data augmentation and unsupervised sample selection. As our objective is to evaluate *cross-lingual adaptation* performance and not *cross-lingual representation*, we train our model with the original source and augmented source and target language data, while validating it with target development data for model selection. We refer this as **zero-resource** setup, which is still a *minimal supervision* setting for task adaptation because no *true* target labels are used for training the model. This setup also gives us a way to compare how far we are from the supervised adaptation setting (train and validate on target language data).

## E.2  USE OF MBERT VS. XLM-R

From Table 4, we see that mBERT Devlin et al. (2019) trains the smallest multi-lingual language model (LM) in terms of training data size and model parameters, while XLM-R is the largest one.

Table 4: Training data size and number of model parameters of Cross-lingual Language Models.

| Model Name | Tokenization | Language | #Head | #Layer | #Representation | #Vocab. | #Params. | Dataset. | Data size. |
|---|---|---|---|---|---|---|---|---|---|
| mBERT | cased | 104 | 12 | 12 | 767 | 110k | 172M | wiki | $\sim 100$ GB |
| mBERT | uncased | 102 | 12 | 12 | 767 | 110k | 172M | wiki | $\sim 100$ GB |
| XLM-15 | uncased | 15 | 8 | 12 | 1024 | 95K | 250M | wiki | $\sim 100$ GB |
| XLM-17 | cased | 17 | 16 | 16 | 1024 | 200k | 570M | wiki | $\sim 100$ GB |
| XLM-100 | cased | 100 | 16 | 16 | 1280 | 200k | 570M | wiki | $\sim 100$ GB |
| XLM-R | cased | 100 | 16 | 16 | 1280 | 200k | 570 | CC-100 | 2.5 TB |

At its heart, XLA uses the generation capability of a pre-trained LM for data augmentation, which could be a bottleneck for XLA's performance. In our initial experiments, we found that the generation quality of mBERT is not as good as that of XLM-R. Using mBERT as the vicinity model can thus generate noisy samples that can propagate to the task models and may thwart us from getting the

maximum benefits from the XLA framework. Thus to ensure the generation of better vicinity samples, we choose to use XLM-R - the best performing multi-lingual LM to date, as the vicinity model $\theta_{lm}$ in our framework.

For the task model $\theta^{(i)}$, in principle we can use any multilingual model (*e.g.,* mBERT, XLM-R) while using XLM-R as the vicinity model. However, if we use a weaker model (*e.g.,* mBERT) compared to the vicinity model, the performance gain may not be easily distinguishable, *i.e.,* the gain may come from the increased generalization capability of the stronger vicinity model. This, in turn, can make us unable to evaluate the XLA framework properly in terms of its adaptation capability. In addition, from Table 1 and Table 2 (in the main paper), we observe that the zero-shot XLM-R outperforms mBERT in the *warm-up* step by $\sim 3.8\%$ in NER and $\sim 13.46\%$ in XNLI. Therefore, we choose to use XLM-R for both the task model $\theta^{(i)}$ and vicinity model $\theta_{lm}$. Using this setup, an improvement over the baseline in XLA strictly indicates the superior performance of the framework.

It is also both attractive and challenging to use a single LM (XLM-R) as the vicinity model $\theta_{lm}$ over different languages. Note that the vicinity model in our framework is a disjoint pre-trained entity whose weights are not trained on any task objective. This disjoint characteristic gives our framework the flexibility to replace $\theta_{lm}$ with a better monolingual LM for a specific target language, which in turn makes our model extendable to utilize stronger and new LMs that may come in future.

### E.3 DATASETS (EXTENDED VERSION)

**XNER.** For XNER, we transfer from English (en) to Spanish (es), German (de), Dutch (nl), Arabic (ar), and Finnish (fi). For English and German, we consider the dataset from CoNLL-2003 shared task Sang and Meulder (2003), while for Spanish and Dutch, we use the dataset from CoNLL-2002 shared task Sang (2002). We collected Arabic and Finnish NER datasets from Bari et al. (2020). The NER tags are converted from IOB1 to IOB2 for standardization and all the tokens of each of the six (6) datasets are classified into five (5) categories: *Person, Organization, Location, Misc.,* and *Other.* Pre-trained LMs like XLM-R generally operate at the subword level. As a result, when the labels are at the word level, if a word is broken into multiple subwords, we mask the prediction of non-first subwords. Table 9 presents the detail statistics of the XNER datasets. We see that the datasets for different languages vary in size. Also the class-distribution is not balanced in these datasets. Therefore, we use the **micro F1 score** as the evaluation metric for XNER.

Table 5: Statistics of training, development and test datasets in different languages for XNER.

| Lang | Train | Dev. | Test | XLMR data | % of en |
|------|-------|------|------|-----------|---------|
| English | 14041 | 3250 | 3453 | 300.8 GB | 100 |
| Spanish | 8323 | 1915 | 1517 | 53.3 GB | $\sim$17.70 |
| Dutch | 15519 | 2821 | 5076 | 29.3 GB | $\sim$9.74 |
| German | 12152 | 2867 | 3005 | 66.6 GB | $\sim$22.14 |
| Arabic | 2166 | 267 | 254 | 28.0 GB | $\sim$9.30 |
| Finnish | 13497 | 986 | 3512 | 54.3 GB | $\sim$18.05 |

**XNLI.** We use the standard XNLI dataset Conneau et al. (2018) which extends the MultiNLI dataset Williams et al. (2018) to 15 languages. For a given pair of sentences, the task is to predict the entailment relationship between the two sentences, *i.e.,* whether the second sentence (*hypothesis*) is an *Entailment*, *Contradiction*, or *Neutral* with respect to the first one (*premise*). For XNLI, we experiment with transferring from English to Spanish (es), German (de), Arabic (ar), Swahili (sw), Hindi (hi), and Urdu (ur). Unlike NER, from Table 6, we see that the dataset sizes are same for all languages. Also the class-distribution is balanced in all the languages. Thus, we use **accuracy** as the evaluation metric for XNLI.

**PAWS-X.** The task of PAWS (Paraphrase Adversaries from Word Scrambling) (Zhang et al., 2019) is to predict whether each pair is a paraphrase or not. PAWS-X contains six typologically distinct languages: French, Spanish, German, Chinese, Japanese, and Korean. For this task, we experiment

Table 6: Statistics of training, development and test datasets in different languages for XNLI.

| Lang | Train | Dev. | Test | XLMR data | % of en |
|------|-------|------|------|-----------|---------|
| English | 392702 | 2490 | 5010 | 300.8 GB | 100 |
| Spanish | 392702 | 2490 | 5010 | 53.3 GB | ∼17.70 |
| German | 392702 | 2490 | 5010 | 66.6 GB | ∼22.14 |
| Arabic | 392702 | 2490 | 5010 | 28.0 GB | ∼9.30 |
| Swahili | 392702 | 2490 | 5010 | 1.5 GB | ∼0.50 |
| Hindi | 392702 | 2490 | 5010 | 20.2 GB | ∼6.72 |
| Urdu | 392702 | 2490 | 5010 | 5.7 GB | ∼1.89 |

with transferring from English to all of these six languages. Table 7 presents the detail statistics of the PAWS-X datasets. Similar to XNLI, we use **accuracy** as the evaluation metric for this task.

Table 7: Statistics of training, development and test datasets in different languages for PAWS-X.

| Lang | Train | Dev. | Test | XLMR data | % of en |
|------|-------|------|------|-----------|---------|
| English | 49401 | 8000 | 8000 | 300.8 GB | 100 |
| French | 49401 | 2490 | 5010 | 56.8 GB | ∼18.88 |
| Spanish | 49401 | 1962 | 1999 | 53.3 GB | ∼17.70 |
| German | 49401 | 1932 | 1967 | 66.6 GB | ∼22.14 |
| Chinese | 49401 | 1984 | 1975 | 63.5 GB | ∼21.11 |
| Japanese | 49401 | 1980 | 1946 | 69.3 GB | ∼23.04 |
| Korean | 49401 | 1965 | 1972 | 54.2 GB | ∼18.02 |

### E.4 SETTINGS (EXTENDED VERSION)

We present the hyperparameter settings for XNER and XNLI tasks for the XLA framework in Table 8. In the *warm-up* step, we train and validate the task models with English data. However, for *cross-lingual adaptation*, we validate (for model selection) our model with the target language development set. We train our model with respect to the number of steps instead of the number of epochs. In the case of a given number of epochs, we convert it to a total number of steps.

For both tasks, we observe that *learning rate* is a crucial hyperparameter. In table 8, *lr-warm-up-steps* refer to the *warmup-step* from triangular learning rate scheduling Smith (2015). This hyperparameter is not to be confused with *Warm-up step* of the XLA framework. In our experiments, *batch-size* is another crucial hyperparameter that can be obtained by multiplying per GPU training batch size with the total number of gradient accumulation steps. We fix the maximum sequence length to 280 for XNER and 128 tokens for XNLI.

For each of the experiments, we report the average score of three task models, $\theta^{(1)}$, $\theta^{(2)}$, $\theta^{(3)}$, which are initialized with different seeds. We perform each of the experiments in a single GPU setup with *float32* precision.

Table 8: Hyperparameter settings for XNER, XNLI, and PAWS-X task.

| Hyperparameter | XNER | | XNLI | | PAWS-X | |
|---|---|---|---|---|---|---|
| | **Warm-up step** | **X-lingual adaptation** | **Warm-up step** | **X-lingual adaptation** | **Warm-up step** | **X-lingual adaptation** |
| **Training-hyperparameters** | | | | | | |
| model-type | xlm-r L | warm-up-ckpt | xlm-r L | warm-up-ckpt | xlm-r L | warm-up-ckpt |
| sampling-factor $\alpha$ | – | 0.7 | – | 0.7 | – | 0.7 |
| drop-out | 0.1 | 0.1 | 0.1 | 0.1 | 0.1 | 0.1 |
| max-seq-length | 280 | 280 | 128 | 128 | 128 | 128 |
| per-gpu-train-batch-size | 4 | 4 | 16 | 16 | 16 | 16 |
| grad-accumulation-steps | 5 | 4 | 2 | 2 | 2 | 2 |
| logging-step | 50 | 50 | 50 | 25 | 50 | 25 |
| learning-rate (lr) | $3e^{-5}$ | $5e^{-6}$ | $1e^{-6}$ | $1e^{-6}$ | $1e^{-6}$ | $1e^{-6}$ |
| lr-warm-up-steps | 200 | 10% of train | 10% of train | 10% of train | 10% of train | 10% of train |
| weight-decay | 0.01 | 0.01 | – | – | – | – |
| adam-epsilon | $1e^{-8}$ | $1e^{-8}$ | $1e^{-8}$ | $1e^{-8}$ | $1e^{-8}$ | $1e^{-8}$ |
| max-grad-norm | 1.0 | 1.0 | 1.0 | 1.0 | 1.0 | 1.0 |
| num-of-train-epochs | – | 1 | – | 1 | – | 1 |
| XLA-epochs | – | 3 | 6 | 3 | 10 | 6 |
| max-steps | 3000 | – | – | – | – | |
| train-data-percentage | 100 | 100 | 5 | 5 | 5 | 5 |
| conf-penalty | True | False | True | False | True | False |
| **Distillation-hyperparameters** | | | | | | |
| #mixture-component | – | . 2 | – | – | – | – |
| posterior-threshold | – | 0.5 | – | – | – | – |
| covariance-type | – | Full | – | – | – | – |
| distilation-factor $\eta$ | – | 80, 100, 100 | – | 50, 80, 100 | – | 80, 90, 80 |
| distillation-type | – | confidence | – | confidence | – | confidence |
| **Augmentation-hyperparameters** | | | | | | |
| do-lower-case | False | False | False | False | - | False |
| aug-type | – | successive-max | – | successive-cross | – | successive-cross |
| aug-percentage $P$ | – | 30 | – | 30 | – | 40 |
| diversification-factor $\delta$ | – | 3 | – | $2\times2$ | – | $2\times2$ |

## F  Results (extended version)

We include detailed results for CoNLL-XNER, XNLI, and PAWS-X datasets to compare with previous literatures. We also provide standard deviations over three different random seeds here.

Table 9: Results in **F1 score** for Cross-lingual Named Entity Recognition (XNER). "x" represents model fails to converge and "-" represents no results were reported for the setup.

| Model | en | es | nl | de | ar | fi |
|---|---|---|---|---|---|---|
| *Supervised Result* | | | | | | |
| (Char+fastText) bi-LSTM-CRF Bari et al. (2020) | $89.77 \pm 0.19$ | $84.71 \pm 0.06$ | $85.16 \pm 0.21$ | $78.14 \pm 0.42$ | $75.49 \pm 0.53$ | $84.21 \pm 0.13$ |
| XLM-R Conneau et al. (2020) | 92.92 | 89.72 | 92.53 | 85.81 | – | – |
| XLM-R (our imp.) | $92.9 \pm 0.23$ | $89.2 \pm 0.37$ | $92.9 \pm 0.21$ | $86.2 \pm 0.32$ | $86.8 \pm 0.53$ | $92.4 \pm 0.2$ |
| *Zero-Resource Baseline* | | | | | | |
| fastText-bi-LSTM-CRF Bari et al. (2020) | $88.98 \pm 0.25$ | x | x | x | x | x |
| (Char+fastText)bi-LSTM-CRF Bari et al. (2020) | $89.92 \pm 0.15$ | $26.76 \pm 1.45$ | $20.94 \pm 0.74$ | $8.34 \pm 1.43$ | x | $22.44 \pm 2.23$ |
| BERT-base-cased | $91.21 \pm 0.18$ | $52.88 \pm 1.33$ | $29.16 \pm 3.30$ | $44.41 \pm 2.36$ | x | $30.18 \pm 1.93$ |
| Keung et al. (2019) | – | 75.00 | 77.50 | 68.60 | – | – |
| Wang et al. (2019) | – | 75.77 | 79.03 | 70.54 | – | – |
| Wu and Dredze (2019) | – | 74.96 | 77.57 | 69.56 | – | – |
| Pires et al. (2019) | – | 73.59 | 77.36 | 69.74 | – | – |
| Conneau et al. (2020) | – | 78.64 | 80.80 | 71.40 | – | – |
| Bari et al. (2020) | – | $75.93 \pm 0.81$ | $74.61 \pm 1.24$ | $65.24 \pm 0.56$ | $36.91 \pm 2.74$ | $53.77 \pm 1.54$ |
| mBERT-cased (Our implementation) | $91.13 \pm 0.14$ | $74.76 \pm 1.06$ | $79.58 \pm 0.38$ | $70.99 \pm 1.24$ | $45.48 \pm 1.47$ | $65.95 \pm 0.76$ |
| XLM-R (Our implementation) | $92.23 \pm 0.19$ | $79.29 \pm 0.43$ | $80.87 \pm 0.90$ | $73.40 \pm 0.96$ | $49.04 \pm 1.19$ | $75.57 \pm 0.94$ |
| XLM-R (ensemble) | 92.76 | 80.62 | 81.46 | 75.4 | 52.3 | 76.85 |
| *Our Method* | | | | | | |
| mBERT-cased + conf-penalty | $90.81 \pm 0.17$ | $75.06 \pm 0.63$ | $79.26 \pm 0.65$ | $72.31 \pm 0.52$ | $47.03 \pm 1.65$ | $66.72 \pm 0.44$ |
| XLM-R + conf-penalty | $92.49 \pm 0.09$ | $80.45 \pm 0.42$ | $81.07 \pm 0.12$ | $73.76 \pm 1.01$ | $49.94 \pm 0.43$ | $76.05 \pm 0.25$ |
| XLA | – | $83.05 \pm 0.38$ | $85.21 \pm 0.23$ | $80.33 \pm 0.07$ | $57.35 \pm 0.56$ | $79.75 \pm 0.34$ |
| XLA (ensemble) | – | **83.24** | **85.32** | **80.99** | **58.29** | **79.87** |

Table 10: Results in **Accuracy** for Cross-lingual Natural Language Inference (XNLI) task.

| Model | en | es | de | ar | sw | hi | ur |
|---|---|---|---|---|---|---|---|
| *Supervised Result (TRANSLATE-TRAIN-ALL)* | | | | | | | |
| Huang et al. (Wiki+MT) Huang et al. (2019) | 85.6 | 82.3 | 80.9 | 78.2 | 73.8 | 73.4 | 69.6 |
| XLM-R (Base) Conneau et al. (2020) | 85.4 | 82.2 | 80.3 | 77.3 | 73.1 | 76.1 | 73.0 |
| XLM-R Conneau et al. (2020) | 89.1 | 86.6 | 85.7 | 83.1 | 78.0 | 81.6 | 78.1 |
| *Zero-Resource Baseline for* **Full (100%) English labeled** *training set* | | | | | | | |
| mBERT-cased Wu and Dredze (2019) | 82.1 | 74.3 | 71.1 | 64.9 | 50.4 | 60.0 | 58.0 |
| XLM Lample and Conneau (2019) | 83.2 | 76.3 | 74.2 | 68.5 | 64.6 | 65.7 | 63.4 |
| XLM-R (Paper) Conneau et al. (2020) | 89.1 | 85.1 | 83.9 | 79.8 | 73.9 | 76.9 | 73.8 |
| XLM-R (XTREME) Hu et al. (2020) | 88.7 | 83.7 | 82.5 | 77.2 | 71.2 | 75.6 | 71.7 |
| XLM-R (Our implementation) | $88.87 \pm 0.31$ | $84.34 \pm 0.37$ | $82.78 \pm 0.56$ | $78.44 \pm 0.50$ | $72.08 \pm 1.05$ | $76.40 \pm 0.87$ | $72.10 \pm 1.22$ |
| XLM-R (ensemble) | 89.24 | 84.73 | 83.27 | 79.06 | 73.17 | 77.23 | 73.07 |
| *Our Method* | | | | | | | |
| XLM-R + conf-penalty | $88.83 \pm 0.12$ | $84.30 \pm 0.24$ | $82.86 \pm 0.14$ | $78.20 \pm 0.38$ | $71.83 \pm 0.41$ | $76.24 \pm 0.47$ | $71.62 \pm 0.70$ |
| XLA | – | $85.65 \pm 0.04$ | $84.18 \pm 0.46$ | $80.50 \pm 0.19$ | $74.70 \pm 0.47$ | $78.84 \pm 0.32$ | $73.35 \pm 0.41$ |
| XLA (ensemble) | – | **86.12** | **84.61** | **80.89** | **74.89** | **78.98** | **73.45** |
| *Zero-Resource Baseline for* **5% English labeled** *training set* | | | | | | | |
| XLM-R (Our implementation) | $83.08 \pm 1.04$ | $78.48 \pm 0.76$ | $77.54 \pm 0.60$ | $72.04 \pm 0.79$ | $67.3 \pm 0.66$ | $70.41 \pm 0.09$ | $66.72 \pm 0.29$ |
| XLM-R (ensemble) | 84.65 | 79.56 | 78.38 | 72.22 | 66.93 | 71.00 | 66.79 |
| *Our Method* | | | | | | | |
| XLM-R + conf-penalty | $84.24 \pm 0.22$ | $79.23 \pm 0.37$ | $78.47 \pm 0.20$ | $72.43 \pm 0.75$ | $67.72 \pm 0.17$ | $71.08 \pm 0.73$ | $67.63 \pm 0.62$ |
| XLA | – | $81.53 \pm 0.11$ | $80.88 \pm 0.28$ | $77.42 \pm 0.15$ | $72.31 \pm 0.12$ | $74.70 \pm 0.26$ | $70.84 \pm 0.22$ |
| XLA (ensemble) | – | **82.35** | **81.93** | **78.56** | **73.53** | **75.20** | **71.15** |

Table 11: Results in **Accuracy** for PAWS-X task.

| Model | en | de | es | fr | ja | ko | zh |
|---|---|---|---|---|---|---|---|
| **Supervised Results** (TRANSLATE-TRAIN-ALL) | | | | | | | |
| XLM-R (our impl.) | $95.8 \pm 0.23$ | $92.5 \pm 0.29$ | $92.8 \pm 0.15$ | $93.5 \pm 0.12$ | $85.5 \pm 0.32$ | $86.6 \pm 0.48$ | $87.6 \pm 0.1$ |
| Zero-Resource Baseline for **Full (100%) English labeled** training set | | | | | | | |
| XLM-R (XTREME) | 94.7 | 89.7 | 90.1 | 90.4 | 78.7 | 79.0 | 82.3 |
| XLM-R (our imp.) | $95.46 \pm 0.36$ | $90.06 \pm 0.59$ | $89.92 \pm 0.54$ | $90.85 \pm 0.71$ | $79.89 \pm 1.17$ | $79.74 \pm 1.47$ | $82.49 \pm 0.82$ |
| XLM-R (ensemble) | 96.10 | 90.75 | 90.55 | 91.80 | 80.55 | 80.70 | 83.45 |
| XLM-R+con-penalty | $95.38 \pm 0.15$ | $90.75 \pm 0.29$ | $90.72 \pm 0.56$ | $91.71 \pm 0.31$ | $81.77 \pm 0.63$ | $82.07 \pm 0.54$ | $84.25 \pm 0.36$ |
| XLA | – | $92.27 \pm 0.75$ | $92.28 \pm 0.16$ | $92.85 \pm 0.35$ | $83.88 \pm 0.49$ | $84.27 \pm 0.23$ | $86.90 \pm 0.35$ |
| XLA (ensemble) | – | **92.55** | **92.35** | **93.35** | **84.30** | **84.35** | **86.95** |
| Zero-Resource Baseline for **5% English labeled** training set | | | | | | | |
| XLM-R (our imp.) | $91.15 \pm 0.98$ | $83.72 \pm 1.64$ | $84.32 \pm 1.76$ | $85.08 \pm 1.29$ | $73.65 \pm 1.03$ | $72.60 \pm 2.04$ | $77.22 \pm 1.22$ |
| XLM-R (ensemble) | 92.05 | 84.05 | 84.65 | 85.75 | 74.30 | 71.95 | 77.50 |
| XLM-R+con-penalty | $91.85 \pm 0.70$ | $86.15 \pm 1.37$ | $86.38 \pm 1.02$ | $85.98 \pm 0.44$ | $76.03 \pm 1.51$ | $75.43 \pm 1.32$ | $79.15 \pm 1.14$ |
| XLA | – | $89.05 \pm 0.85$ | $90.27 \pm 0.38$ | $90.12 \pm 0.28$ | $80.50 \pm 0.73$ | $79.60 \pm 0.43$ | $82.65 \pm 0.44$ |
| XLA (ensemble) | – | **89.25** | **90.85** | **90.25** | **81.15** | **80.15** | **82.90** |

# G  EXAMPLES OF AUGMENTED DATA

We present examples of augmented samples generated by our vicinity model for XNER, XNLI, and PAWS-X datasets in Tables 12, 13, and 14 respectively.

Table 12: Examples of augmented data from XNER dataset.

| English |
|---|
| **Original**: Motor-bike registration rose 32.7 percent in the period . 
 **Augmented**: Motor-bike sales rose 32.7 percent in the US . |
| **Original**: He will be replaced by Eliahu Ben-Elissar , a former Israeli envoy to Egypt and right-wing Likud party politician 
 **Augmented**: He will be led by Eliahu Cohen , a former UN Secretary to Egypt and right-wing opposition party leader . |
| **Original**: Israeli-Syrian peace talks have been deadlocked over the Golan since 1991 despite the previous government 's willingness to make Golan concessions . 
 **Augmented**: The peace talks have been deadlocked over the Golan since 2011, despite the Saudi government 's willingness to make Golan concessions . |

| Spanish |
|---|
| **Original**: En esto de la comida abunda demasiado la patriotería . 
 **Augmented**: En medio de la guerra abunda demasiado la violencia . |
| **Original**: Pero debe , cómo no , estar abierta a incorporaciones foráneas . 
 **Augmented**: También debe , cómo no , estar abierta a personas diferentes . |
| **Original**: Deutsche Telekom calificó esta compra , cuyo precio no especificó , como otro paso hacia su internacionalización mediante adquisiciones mayoritarias destinadas a tener el control de la dirección de esas empresas . 
 **Augmented**: Deutsche Bank calificó esta operación , cuyo importe no especificó , como otro paso hacia su expansión mediante acciones mayoritarias destinadas a tener el control de la dirección de las empresas . |

| Dutch |
|---|
| **Original**: Onvoldoende om een zware straf uit te spreken , luidt het . 
 **Augmented**: Onvoldoende om een zware waarheid uit te leggen , is het . |
| **Original**: Dit hof verbindt nu geen straf aan de schuld die ze vaststelt . 
 **Augmented**: Dit hof geeft nu de schuld aan de schuld die ze vaststelt . |
| **Original**: Wat jaren meeging als een omstreden ' CVP-dossier ' krijgt nu door de rechterlijke uitspraak het cachet van een oude koe in de gracht . 
 **Augmented**: Wat jaren begon als een omstreden ' CVP-dossier ' krijgt nu door de rechterlijke macht het cachet van de heilige koe in de gracht . |

| German |
|---|
| **Original**: Gleichwohl bleibt diese wissenschaftlich abgeleitete Klassifizierung von Erzähltypen nur äußerlich . 
 **Augmented**: Gleichwohl bleibt die daraus abgeleitete Klassifizierung von Erzähltypen nur begrenzt . |
| **Original**: Dies führt vielmehr zu anderen grundlegenden Mißverständnissen , die zur Verwischung entscheidender Unterschiede beitragen . 
 **Augmented**: Dies führt vielmehr zu sehr großen Mißverständnissen , die zur Verwischung entscheidender Informationen führen . |
| **Original**: Die eine Geschichte zerfällt dabei in viele Erzählungen , die wiederum wissenschaftlich genau nach unterschiedlichen Genres klassifiziert werden können . 
 **Augmented**: Die ganze Geschichte zerfällt dabei in viele Erzählungen , die nicht ganz genau in verschiedene Genres gestellt werden können . |

Table 13: Examples of augmented data from XNLI dataset.

| English |
|---|
| **Original**: |
| *text_a*: One of our number will carry out your instructions minutely . |
| *text_b*: A member of my team will execute your orders with immense precision . |
| **Augmented**: |
| *text_a*: One of our number will carry out your order immediately |
| *text_b*: A member of my team will execute your orders with immense care . |
| **Original**: |
| *text_a*: my walkman broke so i 'm upset now i just have to turn the stereo up real loud |
| *text_b*: I 'm upset that my walkman broke and now I have to turn the stereo up really loud . |
| **Augmented**: |
| *text_a*: my stereo broke so i 'm stuck. i just have to turn the stereo up super loud |
| *text_b*: I 'm upset because my phone broke and now I have to turn the music up really loud . |

| Spanish |
|---|
| **Original**: |
| *text_a*: Bueno , porque lo caliente que quiero decir como en el más frío que se pone en invierno ahí abajo , cuánto es ? |
| *text_b*: Hace calor todo el tiempo donde vivo , incluido el invierno . |
| **Augmented**: |
| *text_a*: Bueno , pero lo primero que quiero decir como en el caso calor que se pone en invierno ahí arriba , cuánto es ? |
| *text_b*: Tengo calor todo el tiempo que puedo , incluido el invierno . |
| **Original**: |
| *text_a*: Sí , es como en louisiana donde ese tipo que es como un miembro del ku klux klan algo fue elegido un poco aterrador cuando piensas en eso . |
| *text_b*: Un miembro del ku klux klan ha sido elegido en louisiana . |
| **Augmented**: |
| *text_a*: Sí , estuvieron en louisiana y ese tipo que aparece como un miembro del ku klux klan algo ha sido un poco aterrador cuando piensas en eso . |
| *text_b*: Un miembro del kumite klan ha sido detenido en louisiana . |

| Arabic |
|---|
| **Original**: |
| *text_a*: انه بطيء ، هناك العديد من الالات الافضل في السوق الان |
| *text_b*: هذه اسرع الة ، لن تجدي الة افضل . |
| **Augmented**: |
| *text_a*: انه صحيح ، هناك الكثير من الالات الافضل في السوق الان |
| *text_b*: هذه اسرع طريقة ، لنجعل الة افضل . |
| **Original**: |
| *text_a*: وقد استغرق ذلك ظهور سفن النفاثة وسفن الرحلات البحرية لكي يحدث ذلك . |
| *text_b*: لا توجد سفن سياحية في المنطقة . |
| **Augmented**: |
| *text_a*: وقد سبب ذلك ظهور السيارات النفاثة وسفن الحرب البحرية لكي يحدث ذلك . |
| *text_b*: لا توجد أنشطة أخرى في المنطقة . |

Table 14: Examples of augmented data from PAWS-X dataset.

| **English** |
| --- |

**Original**:
*text_a*: At this time Philips Media was taken over by Infogrames , who became the publisher of the game .
*text_b*: At the time , Infogrames was taken over by Philips Media , who became the publisher of the game .
**Augmented**:
*text_a*: At this time Philips Games was taken over by Ubisoft , which became the developer of the game .
*text_b*: At the end , it was taken over by Philips Software , who became the publisher of the magazine .

**Original**:
*text_a*: In November , 2010 , she was rated as the fifth highest ranked under-20 female player in the world
*text_b*: In November 2010 , she was rated as the fifth highest player in the world .
**Augmented**:
*text_a*: In March , 2010 , she was rated as the second highest ranking professional female player in the world
*text_b*:In March 2015 , she was listed as the second highest player in the world .

| **French** |
| --- |

**Original**:
*text_a*: Vers 1685, il s'installe à Neu - France et réside quelque temps au Québec.
*text_b*: Il s'installe à Québec vers 1685 et réside en Nouvelle-France depuis un certain temps.
**Augmented**:
*text_a*: En 1950 il s'installe à Neu - York et passe quelque temps au Québec.
*text_b*: Il arrive à Paris vers 1960 et vit en Nouvelle-France depuis un certain temps.

**Original**:
*text_a*: Roger Kirk est né à East London. Il a été élevé et éduqué à Norfolk.
*text_b*: Roger Kirk est né à East London. Il a fait ses études et a grandi à Norfolk.
**Augmented**:
*text_a*: William Shakespeare est né à East London. Il a été élevé et educat à Norfolk.
*text_b*: Robert Kirk est né à East. Il a fait ses études et a grandi à Norfolk.

| **Korean** |
| --- |

**Original**:
*text_a*: 그의 시스템은 개인 가정, 시골 지역, 군대 캠프, 많은 병원 및 영국 Raj 지역에서 채택되었습니다.
*text_b*:그의 시스템은 개인 주택, 시골 지역, 군대 캠프, 많은 병원 및 영국 전역에서 광범위하게 채택되었습니다.
**Augmented**:
*text_a*:이 내용은 개인, 시골, 군대 캠프, 많은 병원 및 영국 여러 지역에서 채택되었습니다.
*text_b*:이 정보는 개인, 시골, 군대 캠프, 많은 병원 및 영국 전역에서 많이 채택되었습니다.

**Original**:
*text_a*:불가리 (Bulhar)는 소말리랜드 북서쪽 서북부의 고고학 유적지입니다.
*text_b*:불가리 (Bulhar)는 북서쪽의 소말리랜드 북서쪽에있는 고고학 유적지입니다.
**Augmented**:
*text_a*: 바르 (Bulhar)는 소말리랜드의 서북부의 고고학 유적지입니다.
*text_b*:불가리 (Bulhar)는 북서쪽의 소말리랜드의 세계 유적지입니다.

## H  ACKNOWLEDGEMENT

We would like to thank the authors and contributors of the **Transformers library** (Wolf et al., 2019). We use their implementation for XLM-R language model. We would also like to thank the authors and contributors of the **fairseq library** (Ott et al., 2019). We use their implementation of XLM-R language model from *torch.hub*. We also thank *Alexis Conneau* for his replies in github repositories.

