# OpenReview forum: "XLA: A Robust Unsupervised Data Augmentation Framework for Cross-Lingual NLP"
_ICLR.cc/2021/Conference — Reject_

### Official Review · AnonReviewer2 · 2020-10-21
**Official Blind Review #2**

**Rating:** 5
**Confidence:** 4

**Review:**

Summary: The paper  presents a data augmentation framework for zero-shot cross-lingual transfer learning. The framework uses different types of data (labeled source data, unlabeled source data, automatically generated augmented data) for training a model for the target language. Experiments are conducted on three different tasks: Named Entity Recognition (NER), Natural Language Inference (NLI) and paraphrase identification (PAWS). The approach combines multiple components together namely self-training, augmented sentence generation and confidence penalty.

Data Augmentation:
The paper states that data augmentation has been successfully used in images but not so much in  text (excluding back-translation). In fact, replacing creating augmenting text by masking and replacing word has been used in NLP both before and after pre-trained LMs such as BERT, etc.

- Contextual Augmentation: Data Augmentation by Words with Paradigmatic Relations. Kobayshi. NAACL 2018
- Conditional BERT Contextual Augmentation. Wu et al. 2018.
- AUG-BERT: An Efficient Data Augmentation Algorithm for Text Classification. Shi et al. CSPS 2019
- A lexical and frame-semantic embedding based data augmentation approach to automatic categorization .... Wang and Yang. EMNLP 2015.

The paper argues that generating new samples for data augmentation using the vicinity distribution of the source and target samples is better than back-translation since you cannot transfer the labels in sequence tagging tasks with back translation. However, similar problems would occur with vicinity distributions based augmentation since even changing a single word may result in changing the meaning of the sentence and hence the label (as the paper argues in Section 3.3.) . Given that, it would be useful to see more discussion/experiments comparing the two types of augmentation strategies especially that the self-training step can leverage augmented data without labels

Experiments:
The paper provide a lot of interesting ablation and analysis. However one of the key questions that I couldn't get an answer to is what is the value of each source of data and could they be used in any different way. For example, what happens if we only do data augmentation for the source only, or the target only or by using translation or back-translation, etc.

On a related note, it looks like the specific order in which the datasets are used is important as shown in the experiments but it is not very clear what is the intuition behind that choice and whether other choices were considered or tried.

Other questions:

What is the benefit of Successive cross (vs. successive max)?

Wikiann is much bigger and has covers 40 languages. any reason why only 3 languages were considered from Wikiann and other languages from CoNLL?

 -----
Edited after authors responses:
I would like to thank the authors for the detailed response and the changes they have made to the paper.

- Regarding contextual data augmentation (Kobayshi et al., Wu et al., etc.): Thanks for pointing out that these method use the labeled data to finetune the LM to make sure that words are replaced with other "label-compatible" words. Note that the comparison is not intended to necessarily show that the proposed method outperforms these baselines. Rather it is intended to guide the reader into making a decision about which method is more appropriate for which problem. If the findings are that the performance is comparable but one method will eliminate the additional finetuning step, that would be a useful finding to share. Also , it is not clear that  these methods would require labeled data in the target language for the finetuning or not.  For example, can the source labeled data be used for the finetuning step?

- Regarding translation: Cross-Lingual transfer via Machine Translation does suffer from the label transfer problem for sequence tagging tasks (transferring labels for sentence-level tasks is straightforward). However, there has been several methods to address this in the literature by using unsupervised word alignment  (e.g., Yarowsky et al., 2001; Ni et al., 2017) or attention weights from NMT models (Schuster et al., 2019), heuristic approaches (e.g., Ehrmann et al., 2011) or co-learning alignment and tagging (Xu et al., 2020).

A detailed comparison and discussion of the trade-off between the performance of each method and the resources they require would make the paper much stronger

---

> ### Author Response · Authors · 2020-11-14
> **Comparison with previous work and Back-Translation**
>
> Thanks for your comments and for recognizing our Experiments and Analysis. We address your concerns below:
>
> ### Discussion about previous supervised data-augmentation literature
>
> Although the data augmentation strategy proposed in Contextual Augmentation (Kobayashi, 2018), Conditional BERT (Wu et al. 2018), AUG-BERT (Shi et al. 2019) seem similar to our proposed LM augmentation method, there are some fundamental differences that we would like to highlight.
>
> Each of these approaches is a constrained augmentation method which alters a pretrained language model to a label-conditional language model for a specific task. This language model fine-tuning procedure is supervised and requires the labels of the data. This means their methods update the parameters of the pretrained language models.
>
> On the contrary, our proposed data augmentation method in the XLA framework is unsupervised and we do not perform any fine-tuning of the pretrained language model (i.e., the vicinity model $\theta_{lm}$).
>
> Note that the vicinity model ($\theta_{lm}$) in our framework is a disjoint pre-trained entity whose weights are not trained on any task objective. This disjoint characteristic gives our framework the flexibility to replace $\theta_{lm}$ even with a better monolingual language model for a specific target language, which in turn makes our model extendable to utilize stronger and new language models that may come in the future. We have added this discussion in our revised version (Section 2 and the last paragraph of intro. of Section 3).
>
> ### Comparing with Back-Translation
>
> The traditional back-translation method requires parallel sentences (i.e., supervised), while our XLA framework is unsupervised. By data augmentation in XLA, we mean not only creating new input samples (x) but also generating their labels (y). XLA does it via simultaneous multilingual co-training and a two-stage co-distillation process. On the other hand, back translation as a stand-alone method does not specify a reliable way to get the task labels. Since one is supervised and the other is unsupervised, we believe it is not fair to compare them. In our paper, we however did not claim that generating new samples for data augmentation using the vicinity distribution of the source and target samples is better than back-translation.
>
> Our main goal in XLA is to improve unsupervised zero-resource cross-lingual adaptation through data augmentation. Even to ensure data non-parallelism, we experimented with 5% (random) data setup for XNLI and PAWS-X. Hence, we believe that back-translation is not an appropriate method to compare with our proposed method.

---

> > ### Author Response · Authors · 2020-11-14
> > **Addressing other concerns**
> >
> > ### Concern regarding different types of datasets and their orders in training
> >
> > We utilize different types of data in different epochs (multi-epoch co-teaching).
> > In the first epoch, we use labeled source data and unlabeled target data with pseudo labels ($\mathcal{D}_s$ and $\mathcal{D}'_t$).
> >
> > In epoch 2, we use only virtual target (vicinal) data with pseudo labels ($\tilde{\mathcal{D}}_t$).
> > And finally in epoch 3, we use all the four datasets - $\mathcal{D}_s$, $\mathcal{D}'_t$, $\tilde{\mathcal{D}}_t$, and $\tilde{\mathcal{D}}_s$.
> >
> > In our initial experiments, we did not see any significant improvement with source-only augmentation ($\tilde{\mathcal{D}}_s$) for both mono- and cross-lingual tasks. While pseudo-labeling may induce noise, the model's predictions for in-domain cross-lingual samples are better. Because of this, for a smooth transition, we introduce the vicinal samples after epoch 1. Finally, inspired by the joint training of the cross-lingual language model, in the third epoch we use all four datasets. We also include labeled source data which ensures that our model does not overfit on target distribution as well as persisting the generalization capability of the source distribution.
> >
> > ### Successive Max vs Successive Cross
> >
> > As described in the paper, we use successive max as our core language model augmentation technique; successive cross also uses it. Pairwise tasks like XNLI and PAWS-X have pairwise dependencies: dependencies between a premise and a hypothesis in XNLI or dependencies between a sentence and its possible paraphrase in PAWS-X. To model such dependencies, we use successive cross, which essentially uses cross-product of two successive max applied independently to each part. For XNER, since we have one sequence, we use successive max there. We have added this discussion in our revised version (end of Section 3.2).
> >
> > ### WikiANN related query
> >
> > In our XNER experiments, we primarily use the most popular CoNLL NER dataset which is human-annotated (gold standard) and included in the XGLUE benchmark (Liang et al. 20). WikiANN is a silver standard dataset originally created by Pan et al. (2017) and included in the XTREME benchmark (Hu et al. 20).
> >
> > As our primary CoNLL dataset lacks extremely low-resource and typographically different languages, we select three different languages (Urdu, Bengali, and Myanmar) from the WikiANN dataset which are not only extremely low-resource but also typographically and structurally different. Moreover, these selected languages have different sizes of unlabeled training data: Urdu (ur-20k training samples), Bengali (bn-10K samples), and Burmese (my-100 samples). These settings were chosen to demonstrate the robustness of our method.

---

### Official Review · AnonReviewer1 · 2020-10-28
**A good data augmentation (DA) method for cross-lingual NLP, however, lacks enough discussions and comparisons with other existing DA methods in NLP.**

**Rating:** 6
**Confidence:** 4

**Review:**

This paper improves 1-to-1 cross-lingual NLP by proposing a new data augmentation method. Overall, the paper is clearly written. The proposed method is intuitive to understand and is novel in crafting augmentation texts using masked language model of BERT and relabeling these vicinal examples.

Experiments on cross-lingual NER, NLI and paraphrase detection tasks demonstrates the effectiveness of XLA, outperforming  XLM-R methods by a large margin and setting up new baselines for future cross-lingual models.

However, as the paper focuses on data augmentation in NLP, it should discuss about data augmentation methods in mono-lingual (e.g. English) and cross-lingual NLP (if there exists), pointing out their advantages/disadvantages under the paper's setting.

As the paper crafts examples with and without vicinity separately for the target language, which part contributes more to the improvement of XLA over XLM-R?

Pros:
- The augmentation algorithm is clear and results are promising.

Cons:
- Lacking comparisons and discussions with other data augmentation methods in NLP.
- Hard to infer the contribution of $D_t^\prime$, $\tilde{D_t}$ and $\tilde{D_s}$ separately from Fig.1.

---

> ### Author Response · Authors · 2020-11-14
> **Thank you for your review! Addressing your concerns.**
>
> Thanks for your comments and for recognizing our strong results. We address your concerns below.
>
> ### Question regarding other data augmentation methods in NLP
>
> To our knowledge, XLA is the first to achieve success on cross-lingual augmentation in zero-resource scenarios. We did not come across any previous work on our main contribution - "unsupervised cross-lingual data augmentation". Prior (non-peer-reviewed) method XLDA (Singh et al. 2019) requires supervision (parallel data) like the back-translation method.
>
> As `AnonReviewer2`  and `AnonReviewer3`  suggested, we have added a discussion about the differences between our proposed unsupervised vicinity sample creation method and supervised Contextual Augmentation method (Kobayashi, 2018) and its successors (Section 2 and last paragraph of intro of Section 3 in the revised paper). We reproduce it here for your convenience.
>
> Although the data augmentation strategy proposed in Contextual Augmentation (Kobayashi, 2018), Conditional BERT (Wu et al. 2018), AUG-BERT (Shi et al. 2019) seem similar to our proposed LM augmentation method, there are some fundamental differences that we would like to highlight.
>
> Each of these approaches is a constrained augmentation method which alters a pretrained language model to a label-conditional language model for a specific task. This language model fine-tuning procedure is supervised and requires the labels of the data. This means their methods update the parameters of the pretrained language models.
>
> On the contrary, our proposed data augmentation method in the XLA framework is unsupervised and we do not perform any fine-tuning of the pretrained language model (i.e., the vicinity model $\theta_{lm}$). We think it is not fair to compare our unsupervised method with a supervised method.
>
> Note that the vicinity model ($\theta_{lm}$) in our framework is a disjoint pre-trained entity whose weights are not trained on any task objective. This disjoint characteristic gives our framework the flexibility to replace $\theta_{lm}$ even with a better monolingual language model for a specific target language, which in turn makes our model extendable to utilize stronger and new language models that may come in the future.
>
> ### Contributions of different types of datasets
>
> We utilize different types of data in different epochs (multi-epoch co-teaching).
>
> In the first epoch, we use labeled source data and unlabeled target data with pseudo labels ($\mathcal{D}_s$ and $\mathcal{D}'_t$).
>
> In epoch 2, we use only virtual target (vicinal) data with pseudo labels ($\tilde{\mathcal{D}}_t$).
> And finally in epoch 3, we use all the four datasets - $\mathcal{D}_s$, $\mathcal{D}'_t$, $\tilde{\mathcal{D}}_t$, and $\tilde{\mathcal{D}}_s$.
>
> From Fig 1, we observe that in every epoch, there is a significant boost in F1 scores for each of the languages on the XNER task. In general, we see more improvement in the first epoch compared to the latter epochs. But this is not surprising, as in the first epoch we augment relatively similar data (i.e., source and the original target data) compared to LM generated data. The latter epochs further improve the performance which exhibits the cumulative contributions of different data sources in XLA.

---

### Official Review · AnonReviewer3 · 2020-10-29
**Important problem, strong results**

**Rating:** 6
**Confidence:** 3

**Review:**

The authors present an unsupervised data augmentation framework for cross-lingual NLP. Their method, called XLA, combines self-learning with co-learning and filtering. They generate additional synthetic examples by replacing words with predictions from pretrained multilingual masked LM (taken from XLM Conneau 2020). The key contribution in this paper is how they get reliable labels by simultaneously co-training three student models and using them to filter examples for training each other to avoid confirmation bias.

Strengths:
The authors present three representative cross-lingual transfer tasks: XNER, XNLI, and PAWS-X with strong results for all three. For XNER XLA gets SoTA in all languages, and they show higher scores for structurally dissimilar and low-resource languages.

Weaknesses:
The method is quite elaborate requiring three epochs, with different training sets at each step, three different models for labelling, and then a combination of co-teaching, co-distillation and "co-guessing". I think that the justification for all these design decisions is not entirely convincing and it feels slightly over-engineered. How robust is this framework? How generalisable?

I also think that the explanation of the architecture is not as clear as it could be. The text in the intro to Section 3 is augmented with an Algorithm and a diagram in the appendix. Each of these explanations are presented separately, and neither the algorithm not the figure are fully explained. I think the algorithm should go in the appendix and be fully explained. I think the figure should go in the main article and be referred to in the text when describing the model at a high level.

Even with these limitations, I think this paper makes enough of a contribution to warrant being accepted. Hopefully some of my concerns can be addressed before the camera ready paper.

Table 1: Ensemble of what three models?

Why not report the supervised results ie. Like the Conneau et al 2020 paper? I know the main point is the zero-shot performance, but it is an interesting data point anyway especially as you have already included the baseline results for this.

Figure 3: Needs to be described - how did you plot this graph? And how is the reader supposed to see that the right hand plot is more robust to OOD data??

The vicinal risk minimization principle takes up a lot of space and adds quite a bit of complexity to the description of the model. I think you could leave it out and improve the paper - this kind of data augmentation is being done already:
	- Sosuke Kobayashi. 2018. Contextual augmentation:Data augmentation by words with paradigmatic re-lations. InProceedings of the 2018 Conference ofthe North American Chapter of the Association forComputational Linguistics: Human Language Tech-nologies, Volume 2 (Short Papers), pages 452–457,New Orleans, Louisiana. Association for Computa-tional Linguistics

The distillation by clustering method is not fully leveraged as extra information is not present in the experiments. It would be a nice extra contribution to the paper if features such as sentence length were indeed included. As it is you might as well leave clustering out of the paper and use the space for something else.

Your appendix is extremely large. Please consider what is truly important and remove the rest.

------

Response to author's replies:
I am impressed by the detailed response and the changes they have made to their paper and I am happy for this paper to be accepted. I still feel like the XLA framework is quite involved and it would have been good to understand which components of this framework are crucial to its success which is why I do not increase my score.

---

> ### Author Response · Authors · 2020-11-14
> **Justification for the design principles of XLA framework**
>
> Thanks for your comments and recognizing our contributions. We address your concerns below and revised our paper accordingly.
>
> Here are our justifications for various design principles of the XLA framework.
>
> ### Is using three models with different initialization necessary?
>
> Yes, different initialization ensures different convergence paths, which results in diversity during inference. Co-labeling (Section 3.3) utilizes this property. There could be some other ways to achieve the same thing. For example, our initial attempt with three different heads (sharing a backbone net) didn't work well.
>
> ### Is using three epochs necessary?
>
> We utilize different types of datasets in different epochs. While pseudo-labeling may induce noise, the model's predictions for in-domain cross-lingual samples are usually better. Because of this, for a smooth transition, we apply the vicinal samples in the second epoch. Finally, inspired by the joint training of the cross-lingual language model,  in the third epoch we use all four datasets. We also include the labeled source data which ensures that our model does not overfit on target distribution as well as persists the generalization capability of the source distribution.
>
> ### Need for the combination of co-teaching, co-distillation and co-guessing?
>
> The combination of these helps to distill out the noisy samples better.
>
> ### Efficiency of the method and expensive extra costs for large-scale pretrained models
>
> It is a common practice in model selection to train 3-5 disjoint LM-based task models (e.g., XLM-R on NER) with different random seeds and report the ensemble score or score of the best (validation set) model. In contrast, XLA uses 3 different models and jointly trains them where the models assist each other through distillation and co-labeling. In that sense, the extra cost comes from distillation and co-labeling, which is not significant and is compensated by the significant improvements that XLA offers.
>
> We added this discussion in Appendix A of our revised version.

---

> > ### Author Response · Authors · 2020-11-14
> > **Addressing other concerns**
> >
> > ### Robustness of XLA framework
> >
> > We have added an additional experiment in the revised version of our paper to show the robustness of XLA (Section 5.3). From Table 6 results we see that after finetuning on a specific target language, the F1 scores of English remain almost similar. This indicates that the augmentation of XLA does not overfit the target language.
> >
> > ### Generalizability of XLA framework
> >
> > We validate the effectiveness of XLA by performing extensive experiments on three different zero-resource cross-lingual transfer tasks – XNER, XNLI, and PAWS-X, which posit different sets of challenges. For example, XNER is a token labeling task, while XNLI and PAWS-X are sequence classification tasks. While PAWS-X requires to classify the adversarial examples for paraphrase identification with high lexical overlap (example: "flights from NY to Florida" and "flights from Florida to NY" are not paraphrase), XNLI requires high-level causal understanding for the classification. In addition to these, we also have shown the effectiveness of our framework on high/low resource, typologically, and structurally different languages (as you have mentioned).
> >
> > To understand the effectiveness and generalizability even better, we are also working on in-domain data augmentation for low-resource machine translation using the XLA framework. Our preliminary results are quite promising.
> >
> > ### Query regarding ensembling:
> >
> > In our algorithm, we apply simultaneous self-training on three models with different seeds ($\theta^{(1)}$, $\theta^{(2)}$, $\theta^{(3)}$).  We take the average probabilities from these three models for reporting the ensemble results (Section 4.2).
> >
> > ### Comparison with supervised results
> > We have added the supervised results  (translate-train-all)  of XNER, XNLI, and PAWS-X in the revised version (Table 1,3,4 respectively).
> >
> > One interesting fact on CoNLL NER tasks is that our unsupervised zero-resource setup exhibits very competitive results and sometimes outperforms the supervised bi-LSTM-CRF model trained with fastText embeddings in Es, De, and  NL  (Table 1). Moreover, surprisingly on PAWS-X task, our XLA (ensemble) achieves almost similar accuracies compared to supervised finetuning of XLM-R on all target language training datasets.
> >
> > ### Query regarding Figure 3
> >
> > As described in Section C.1, Figure 3 presents the effect of the confidence penalty (Eq. 1 in the main paper) for zero-shot transfer; in this case the predictions of the XLM-R model (trained on English NER) on the Spanish dev. data. We show the class distribution (after applying Softmax) for the Spanish examples using t-SNE. From the visualization, it can be seen that the model trained with the confidence penalty shows better inter-class separation which exhibits robustness of the multilingual model.
> >
> > We have revised the caption (Appendix C) in the revised paper.
> >
> > ### Comparing XLA’s data augmentation with Kobayashi et al.
> >
> > Although the data augmentation strategy proposed in the Contextual Augmentation method (Kobayashi, 2018) seems similar to XLA's LM augmentation, there are some fundamental differences between these two approaches.
> > The Contextual Augmentation method is a constrained augmentation method which converts a pretrained language model to a label-conditional language model for a specific task. This language model fine-tuning procedure is supervised and requires task labels for the data. This means that their method updates the parameters of the pretrained language model.
> > On the contrary, our proposed data augmentation method in the XLA framework is unsupervised and we do not perform any fine-tuning of the pretrained language model (i.e., the vicinity model $\theta_{lm}$). Note that the vicinity model ($\theta_{lm}$) in our framework is a disjoint pre-trained entity whose weights are not trained on any task objective. This disjoint characteristic gives our framework the flexibility to replace $\theta_{lm}$ even with a better monolingual language model for a specific target language, which in turn makes our model extendable to utilize stronger and new language models that may come in the future.
> >
> > We have added this discussion in our revised version (Section 2 and last paragraph of intro. of Section 3) .
> >
> > ### Regarding Figures and Text
> >
> > Given the space limitations, we had to choose between the Algorithm and the Figure. We thought the algorithm is more informative in this case, so we kept it in the main content and put the figure in the Appendix. Thanks for your suggestion. We have brought the figure from the appendix to the main paper in the revised version.
> >
> > We have also added an additional intro. in appendix so that the readers can explore the content of the appendix more easily.

---

### Official Review · AnonReviewer4 · 2020-10-30
**Good experiment results yet limited novelty**

**Rating:** 5
**Confidence:** 3

**Review:**

This paper proposed a new data augmentation framework for low-resourse (and zero resource) cross-lingual task adaptation by combining several methods (entropy regluarized training, self-training). The authors conducted extensive experiments on three cross-lingual tasks, demonstrating the effectiveness of XLA. In addition, the authors compared different choices in the XLA distilation stage and claimed that the gain from XLA is beyond the model ensemble effect.

According to the experiment results, the XLA framework has a remarkable gain over previous methods. However, it's not clear which component actually contributed to this gain. An thorough ablation study on different model component will help clarify this concern.

The writing style is a bit redundant and confusing. For instance, the author use "model distillation" to define the label selection procedure, yet, this term is another technique in machine learning literature.

Lastly from a methodology perspective, I think the algorithm seems like a combination of existing methods and its novelty is incremental to me. I hope the authors can edit the paper more carefully with a clarification on their novelty.

---

> ### Author Response · Authors · 2020-11-14
> **Thank you for your review! Addressing your concerns.**
>
> Thanks for your comments and recognizing our strong results. We address your concerns below:
>
> ### Ablation Study
>
> XLA contains the following components: (i) confidence penalty, (ii) two-stage distillation, (iii) different types of augmentation. Our proposed method combines all these components in a unified framework. We have shown a detailed analysis of the components in Section 5. We also have shown additional visualization of the confidence penalty's contribution in appendix C. Moreover, in our revised paper we have added robustness analysis of XLA (Section 5.3).
>
> ### Sample distillation vs. Model distillation
>
> Quoting directly from Section 3.3 of our paper - "We propose two stages of distillation to filter out noisy augmented samples". In our paper, we use the term "distillation" for filtering out the noisy augmented samples. We do not perform any "model distillation" which is popularly known as knowledge distillation in machine learning literature, rather we use our model to distill correctly predicted samples. Thanks for your suggestion. We have addressed this issue in our revised version (Section 3.2).
>
> ### Regarding Novelty
>
> We did not come across any previous work on our main contribution - "unsupervised data augmentation for cross-lingual NLP", which we believe is a significant contribution with the potential of having a broader impact and becoming a generic framework for cross-lingual task adaptation. To the best of our knowledge, we are the first to achieve such success on this problem.
>
> We address the cross-lingual task adaptation problem by successfully integrating simultaneous self-training with unsupervised data augmentation (in 3 folds), an additional loss term (confidence penalty), and sample selection (in 2 folds with 2 different approaches) in a zero-resource setup (without any supervision). Furthermore, our data samples manipulation strategy in mixing different types of data (Section 3.4) is also innovative. We believe such integration in an effective way is non-trivial and a significant discovery, especially considering its remarkable performance across various tasks (as you have mentioned). Also, we’d like to request you to see our response to AnonReviewer3 on why our method’s efficiency is comparable to standard practice which generally uses multiple task models with different random seeds for model selection. We believe XLA can be a generic framework for cross-lingual task adaptation, and it is more important now with the influx of interests in multi-lingual Language Models (e.g., XLM-R, mBERT) and tasks (e.g., XNLI, XNER, PAWS-X, XQuAD, MLQA, MLDoc, etc.).

---

> > ### Comment · Area_Chair1 · 2020-11-14
> > **Please elaborate on novelty**
> >
> > I’d be curious to hear the authors elaborate a bit on the novelty issue. Clearly, previous work has used “unsupervised data augmentation for cross-lingual NLP” with success. Examples range from Täckström et al. (2012) [https://www.aclweb.org/anthology/N12-1052/] to Maharana and Bansal (2020) [https://www.aclweb.org/anthology/2020.findings-emnlp.333.pdf]. Self-training with confidence-based selection, EM, and/or more or less sophisticated sample selection strategies are also part of the standard unsupervised cross-lingual adaptation toolbox. I’m not saying putting all of these together and analysing their effectiveness in the scenarios considered here is not sufficient for a good research paper, but since you claim novelty, which parts, exactly, do you feel bring the most to the table?

---

> > > ### Author Response · Authors · 2020-11-16
> > > **Discussion on Novelty**
> > >
> > > After careful consideration of the paper mentioned in the comment, here is what we found about the approaches in the respective papers.
> > >
> > > `Täckström et al. (2012)`  use dual decomposition aligner from DeNero and Macherey (2011) for the alignment of the cross-lingual word clusters. They use **parallel data** for enforcing constraints on cross-lingual word clusters. They use these cross-lingual word cluster features for transferring linguistic structure from English to other languages. They **did not** do any data augmentation, rather they use cross-lingual word cluster features for knowledge transfer from English to other languages (See Table 5 and 6 of their paper). Although their method should not be compared with any LM based adaptation models, because of the massive pretraining advantage of LM, we would like to point out the performance gap between their method and ours.  For XNER, the baseline we considered is ahead of 26.5% on average to Täckström et al. (2012) (See Table 1 of our paper and Table 6 of their paper).
> > >
> > > Although the search for the best adversarial policy proposed in `Maharana and Bansal (EMNLP-2020, findings)`  is interesting, there are many key differences. First, their set up is TRANSLATE-TEST, where the non-English language samples are first translated to English using a **supervised** MT system (Section 4.6 of their paper). The translated samples are then used in an English monolingual model. So, their setup is **not** zero-resource transfer (relies on a supervised MT system). Second, their data-augmentation (positive and negative perturbations) method is only for monolingual (English).
> > >
> > > While this is not our intention to undermine such great work, we do not find any of the mentioned papers working on our main contribution - "unsupervised data augmentation for cross-lingual NLP".
> > >
> > > Our work comprises (i) conf. penalty,  (ii) co-teaching, (iii) vicinity-based multilingual data augmentation and (iv) reliable sample distillation methods effectively which have not been explored in cross-lingual NLP though they exist in various Machine Learning literature. For example, conf. penalty has been shown to be an effective output regularization method mostly in speech and computer vision tasks. We believe the fact that it also helps large pre-trained masked language models with Transformer architecture in cross-lingual tasks is an interesting finding that may require even future investigation. Similarly, even though the idea of co-teaching has been explored in other NLP tasks, e.g., Ruder and Plank (2018) (https://www.aclweb.org/anthology/P18-1096.pdf), our work is the first to show that they are very effective in cross-lingual adaptation of large multilingual LMs. To our knowledge, both of our vicinity-based multilingual (unsupervised) data augmentation and clustering-based sample distillation methods are novel contributions in NLP.
> > >
> > > It is to be noted that we do not claim novelty of these methods, rather the novelty is claimed for the effective integration of the combination (the XLA framework) of these methods used for cross-lingual augmentation under the zero-resource scenario. With extensive experiments and analysis, we have shown that the effective combination of these ideas give impressive gains across different tasks.

---

### Decision · Program_Chairs · 2021-01-07
**Final Decision**

**Decision:**

Reject

**Comment:**

The paper clearly has merits, presenting a reasonable approach to zero-shot cross-lingual learning with good results, but with limited novelty, perhaps. I am sympathetic to the departure from XTREME on NER, agreeing with the authors that using CoNLL data is more interesting than WikiANN.

The post-rebuttal discussion centered on novelty and baselining - and specifically, whether other approaches to unsupervised data augmentation exist that should be used to baseline the proposed work. The authors argued that most of the approaches mentioned by the reviewers were in some way supervised. I personally think the confusion is a result of the paper being somewhat poorly framed:

Reviewer 2, for example, suggests a bunch of baselines. Some of these require gold labels for supervised fine-tuning to condition the MLM, but this seems like a trivial difference, which is orthogonal to using the augmentation strategies as baselines? Also, other papers have been presented that do not require gold labels, e.g. https://www.aclweb.org/anthology/D18-1100.pdf

Also, on the discussion of Täckström et al. (2012): Older approaches relying on distributional clusters *are* in fact data augmentation methods. Training on augmented data with words replaced is, in the limit, equivalent to training with clusters, when replacement words are sampled from clusters. Others have in the past proposed to use FSAs or clusters induced from static embeddings.

What the authors suggest is a form of co-training procedure, so similarly, semi-supervised algorithms - e.g., tri-training - could have been used as baselines.

In sum, I think the sentiment shared across the reviewers is that the results are largely unsurprising, and could likely be obtained in different ways, including jointly training with a target language modeling objective, tri-training, etc. Finally, I agree with Reviewer 2 that a “detailed comparison and discussion of the trade-off” between the different approaches to data augmentation, even beyond what’s apples-to-apples, would benefit the paper. Maybe there's other advantages to the proposed approach over other baselines (effectiveness, robustness)?